# Asymmetric neurogenic commitment of retinal progenitors involves Notch through the endocytic pathway

**Elisa Nerli[1,2], Mauricio Rocha-Martins[1,2]\*, Caren Norden[1,2]\***

[1]Max Planck Institute of Molecular Cell Biology and Genetics, Dresden, Germany;
[2]Instituto Gulbenkian de Ciência, Oeiras, Portugal

**Abstract** During brain development, progenitor cells need to balanceproliferation and differentiation in order to generate different neurons in the correct numbers and proportions. Currently, the patterns of multipotent progenitor divisions that lead to neurogenic entry and the factors that regulate them are not fully understood. We here use the zebrafish retina to address this gap, exploiting its suitability for quantitative live-imaging. We show that early neurogenic progenitors arise from asymmetric divisions. Notch regulates this asymmetry, as when inhibited, symmetric divisions producing two neurogenic progenitors occur. Surprisingly however, Notch does not act through an apicobasal activity gradient as previously suggested, but through asymmetric inheritance of Sara-positive endosomes. Further, the resulting neurogenic progenitors show cell biological features different from multipotent progenitors, raising the possibility that an intermediate progenitor state exists in the retina. Our study thus reveals new insights into the regulation of proliferative and differentiative events during central nervous system development.

**\*For correspondence:**
mrmartins@igc.gulbenkian.pt (MR-M);
cnorden@igc.gulbenkian.pt (CN)

**Competing interests:** The authors declare that no competing interests exist.

## Introduction

During central nervous system (CNS) development in vertebrates, the balance between proliferation and differentiation is crucial to generate the different types of neurons in the correct numbers and proportions. This in turn ensures functionality of the neural tissue as it is the prerequisite for correct neuronal connectivity and circuit formation. Multipotent progenitors are key to maintain this balance. They first increase in number through a phase of active proliferation (*Fish et al., 2006*; *Homem et al., 2015*) which is followed by neurogenesis onset. At neurogenesis onset, subsets of progenitors leave the cell cycle and differentiate into neurons (*Götz and Huttner, 2005*; *Agathocleous and Harris, 2009*). Expectedly, impairment of the intricate balance between proliferation and differentiation can lead to severe defects of brain development and is associated with numerous human diseases, including microcephaly and tumour formation (*Fish et al., 2006*; *Thornton and Woods, 2009*). Thus, understanding the parameters that differ between cells primed for continued proliferation versus cells that enter the neurogenic program is of utmost importance to comprehend brain development in health and disease.

Previous studies using the mammalian neocortex showed that at the onset of neurogenesis multipotent progenitors start dividing asymmetrically, generating one radial glial cell and one neuron or a basal progenitor and thereby balancing the ratio between differentiation and proliferation at early neurogenesis stages (*Chenn and McConnell, 1995*; *Huttner and Brand, 1997*; *Götz and Huttner, 2005*; *Shen et al., 2006*; *Shitamukai and Matsuzaki, 2012*). Once a sufficient number of progenitors is produced, final symmetric consumptive neuronal divisions deplete the progenitor pool (*Miyata et al., 2004*; *Noctor et al., 2004*). Similarly to the mammalian cortex, asymmetric divisions have also been shown to be important in the fore- and hindbrain of the teleost *Danio rerio* (zebrafish) to ensure the maintenance of the progenitor pool before its final depletion (*Alexandre et al.,*

2010; *Dong et al., 2012*). Thus, while some questions remain, the division modes responsible for the generation of the first neurons in these parts of the nervous system are relatively well explored. What is much less understood, however, is how neurogenic progenitors arise from the multipotent progenitor pool in the first place. Especially in the retina, the tissue that upon maturation enables the perception of visual information, this question has not yet gathered sufficient attention. Here, division patterns have mainly been investigated at later stages when the emergence of the seven different neurons that ensure retinal functionality has already commenced (*Poggi et al., 2005*; *He et al., 2012*; *Kechad et al., 2012*; *Suzuki et al., 2013*). It was shown that these late neurogenic divisions can be asymmetric, generating two neurons of different fate. However, how multipotent progenitors enter the neurogenic path in the first place and what type of division patterns and regulatory pathways these initial fate decisions entail is not yet well explored.

In many parts of the CNS, including the retina, asymmetric divisions leading to the formation of neurons have often been linked to the asymmetric inheritance of distinct cellular components. Examples of differentially inherited components are the basal process (*Saito et al., 2003*; *Okamoto et al., 2013*), the apical domain (*Bultje et al., 2009*; *Alexandre et al., 2010*) or polarity components (*Paolini et al., 2015*; *Zhao et al., 2020*). In addition, signalling pathways including Wnt (*Hirabayashi et al., 2004*) and very prominently Notch (*Dorsky et al., 1995*; *Dorsky et al., 1997*; *Cayouette et al., 2001*; *Ohtsuka et al., 2001*; *Dong et al., 2012*) have been shown to be involved. Notch, for example, can regulate the asymmetry of fate decisions through the asymmetric inheritance of Notch signalling endosomes, as shown in the zebrafish spinal cord and forebrain (*Kressmann et al., 2015*; *Zhao et al., 2020*). In the vertebrate retina, Notch signalling maintains progenitors in a proliferative state through lateral inhibition (*Henrique et al., 1997*; *Jadhav et al., 2006*). It was previously suggested that here Notch regulates neurogenic commitment of multipotent progenitors via an extracellular gradient along the apicobasal axis of the elongated neuroepithelial cells (*Murciano et al., 2002*; *Del Bene et al., 2008*). It was postulated that when nuclei travel along the apicobasal axis during the cell cycle (a phenomenon termed interkinetic nuclear migration, IKNM [*Sauer, 1935*; *Baye and Link, 2007*; *Norden et al., 2009*]), they are exposed to different Notch signalling levels, that in turn influence the progenitors' probability to enter the neurogenic program (*Murciano et al., 2002*; *Baye and Link, 2007*; *Del Bene et al., 2008*). However, the division patterns of retinal progenitors that lead to neurogenic entry and their possible links to Notch signalling were not fully explored.

We here investigate the division patterns that give rise to the neurogenic progenitors, one cell cycle before the generation of post-mitotic neurons. We find that divisions generating neurogenic progenitors are almost always asymmetric and that sister cells do not enter the same lineage. While Notch is involved in the generation of this asymmetry, it does not act via an apicobasal activity gradient. Instead, we provide evidence that Notch signalling acts through the asymmetric inheritance of Sara-positive endosomes. We further show that these first neurogenic progenitors differ in their cell biological features from multipotent progenitors, indicating that they are an intermediate cell type *en route* to differentiation.

## Results

### Ath5 positive neurogenic progenitors arise from asymmetric divisions

To understand how neurogenic progenitors emerge from multipotent progenitors in the developing zebrafish retina, we asked how the earliest of these neurogenic progenitors arise. Possible division patterns were symmetric, asymmetric or a mixture of the two. To follow these earliest neurogenic progenitors, we made use of the well-established fact that they express the transcription factor *ath5* (*atoh7*) (*Poggi et al., 2005*; *Zolessi et al., 2006*) and traced their origin by mosaically labelling cells in zebrafish embryos with an Ath5-driven reporter construct (*Icha et al., 2016a*). We followed the onset of the Ath5 signal using light-sheet imaging from the beginning of neurogenesis at 28 hpf (hours post-fertilisation; *Hu and Easter, 1999*; *Schmitt and Dowling, 1999*; *Martinez-Morales et al., 2005*) as previously described (*Icha et al., 2016b*). We backtracked progenitors that began to express Ath5 (from here on referred to as 'Ath5 positive progenitors'; *Figure 1A*, *Figure 1—figure supplement 1A*) by labelling the plasma membrane and/or the nucleus of the same cells using hsp70:mKate2-ras or hsp70:H2B-RFP constructs (*Figure 1B*). Sister cells of divisions giving

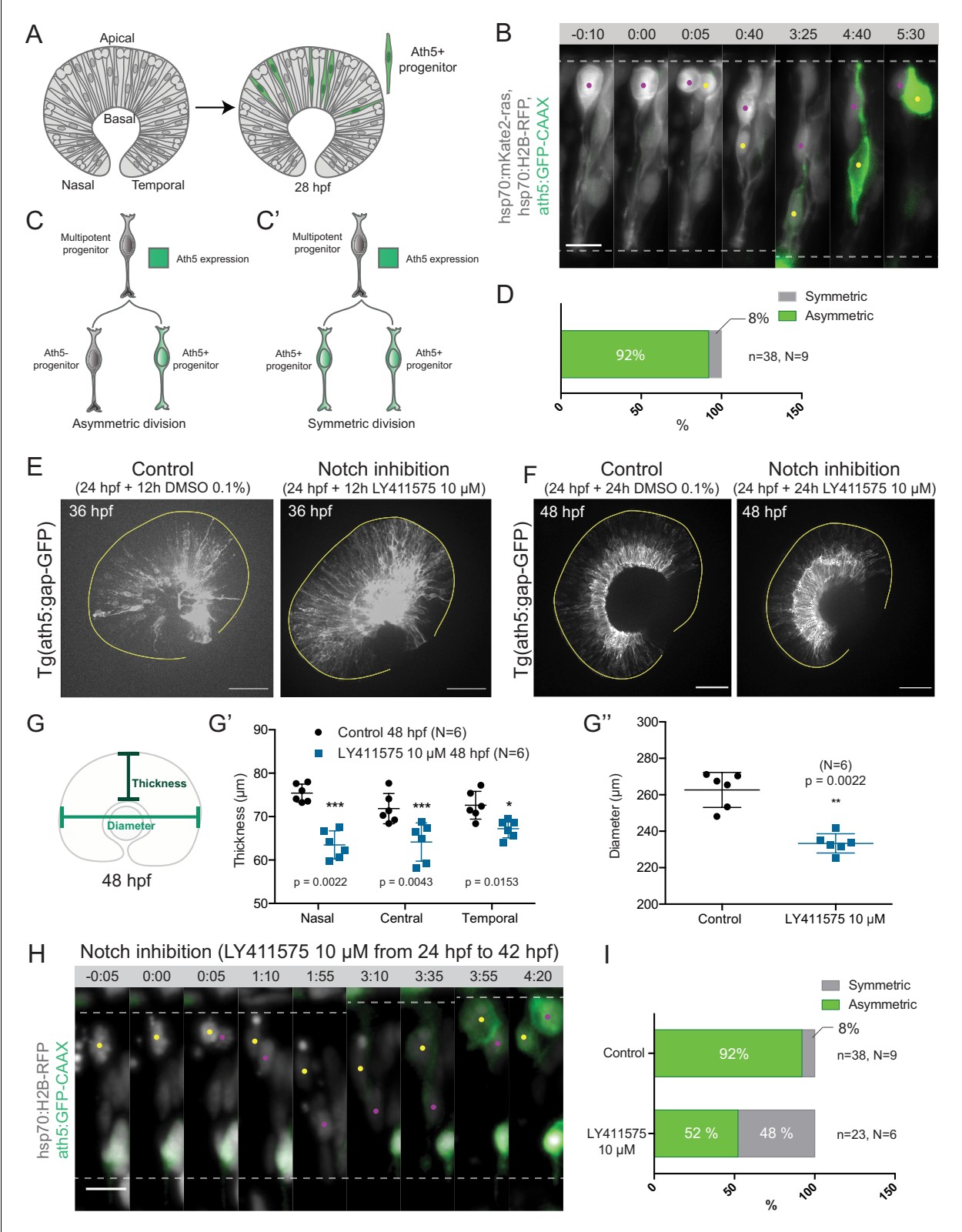

**Figure 1.** Asymmetric cell divisions generate Ath5+ progenitors in a Notch-dependent manner. (**A**) Schematic of mosaic Ath5 expression in the retina (green) at 28 hpf. Injection of ath5:GFP-CAAX construct at 1 cell stage. (**B**) Example of an asymmetric multipotent progenitor division with regards to Ath5 expression onset. hsp70:H2B-RFP (nuclei, grey), hsp70:mkate2-ras (cell membrane, grey), ath5:GFP-CAAX (Ath5, green). Dashed lines show apical and basal sides of the retinal neuroepithelium. Scale bar, 10 µm. Magenta and yellow dots label sister cells. (**C–C'**) Schematics of multipotent

*Figure 1 continued on next page*

*Figure 1 continued*

progenitor cells dividing asymmetrically (C) or symmetrically (C') with regards to Ath5 expression. (D) Distribution of asymmetric vs symmetric divisions observed in live imaging experiments. N = number of embryos, n = number of divisions. (E) Ath5+ cells (grey) at 36 hpf in control (left) vs Notch inhibition (right). Scale bar, 50 μm. The yellow line delimits the apical side of the retinal neuroepithelium. (F) Ath5+ cells (grey) at 48 hpf in control (left) vs Notch inhibition (right). Scale bar, 50 μm. The yellow line delimits the apical side of the retinal neuroepithelium. (G) Schematic of retinal neuroepithelium measurements at 48 hpf, (G') retinal thickness control vs Notch inhibition, (G'') retinal diameter control vs Notch inhibition. Mann-Whitney test used for comparison. Vertical bars represent standard deviation. (H) Example of symmetric progenitor division upon Notch inhibition. hsp70:H2B-RFP (nuclei, grey), ath5:GFP-CAAX (Ath5, green). Dashed lines show apical and basal sides of the retinal neuroepithelium. Scale bar, 10 μm. Magenta and yellow dots label sister cells. (I) Distribution of asymmetric vs symmetric divisions observed in live imaging experiments in Notch inhibition compared to controls.

The online version of this article includes the following video, source data, and figure supplement(s) for figure 1:

**Source data 1.** Source data for panels G',G''.
**Figure supplement 1.** Ath5- sister cells do not enter the Ath5 lineage in the next cell cycle.
**Figure 1—video 1.** Asymmetric division of multipotent retinal progenitors.
https://elifesciences.org/articles/60462#fig1video1
**Figure 1—video 2.** Symmetric division of a multipotent progenitor upon Notch inhibition with 10 μM LY411575, generating two Ath5+ progenitors.
https://elifesciences.org/articles/60462#fig1video2

---

rise to at least one Ath5 positive neurogenic progenitor were backtracked and their division pattern was analysed (*Figure 1C,C'*).

Backtracking of 38 multipotent progenitor divisions from nine embryos (*Figure 1B*, *Figure 1—video 1*) revealed that 92% of these divisions were asymmetric, generating one Ath5 positive (Ath5+) progenitor and one Ath5 negative (Ath5-) progenitor (*Figure 1C,C',D*). Only 8% of divisions were symmetric, producing two Ath5+ cells (*Figure 1C',D*). Following Ath5- sister cells for another cell cycle showed that all cells divided again, but none of the daughter cells started expressing Ath5 (n = 5 divisions, N = 2 embryos; *Figure 1—figure supplement 1B,C*, *Figure 1—video 1*). This suggests that Ath5- sister cells enter a different lineage than their Ath5+ counterpart. Together, these data provide first direct evidence that neurogenic progenitors in the retina emerge from asymmetric divisions, indicating that cellular asymmetry plays an important role in the selection of progenitors that enter the neurogenic program.

## Inhibition of Notch signalling affects the pattern of divisions generating Ath5 positive progenitors

To understand the asymmetry of divisions producing Ath5+ progenitors at the molecular level, we set out to investigate the signalling factors involved. One promising candidate that drives asymmetric divisions in many neurogenic lineages is the Notch pathway (*Bultje et al., 2009*; *Egger et al., 2010*; *Kechad et al., 2012*; *Kressmann et al., 2015*). In the retina, Notch signalling has been shown to maintain progenitors in the proliferative state thus inhibiting neurogenesis (*Bao and Cepko, 1997*; *Jadhav et al., 2006*; *Kechad et al., 2012*; *Maurer et al., 2014*; *Sato et al., 2016*). We therefore tested whether and how Notch depletion influenced the asymmetry of divisions generating Ath5+ progenitors using two established Notch inhibitors, the γ-secretase inhibitors LY411575 (*Figure 1E,F*) and DAPT (*Figure 1—figure supplement 1D*; *Ninov et al., 2012*; *MacDonald et al., 2015*). In line with the reports that Notch signalling suppresses neurogenesis and thereby Ath5 expression onset (*Maurer et al., 2014*), we found that both Notch inhibitors led to an increase in the number of Ath5+ cells at 36 and 48 hours post fertilisation (hpf) (*Figure 1E,F*). LY411575 treatment further led to a decrease in the number of EdU-positive proliferative cells (*Figure 1—figure supplement 1E,F*). The increase of Ath5+ cells was accompanied by a striking reduction in tissue thickness (*Figure 1G,G'*) and retinal diameter (*Figure 1G,G''*) compared to controls at 48 hpf. This retinal size reduction showed that indeed the balance between proliferation and differentiation in the retina was disturbed upon Notch inhibition, leading to tissue-wide consequences.

The increase in the number of Ath5+ progenitors observed upon Notch inhibition could arise from two not mutually exclusive scenarios: Ath5+ progenitors could emerge prematurely by asymmetric divisions that absorb multipotent progenitors too early, or an increase in symmetric divisions giving rise to two Ath5+ progenitors could occur that would result in less progenitors entering another cell cycle, thereby also depleting the pool of remaining progenitors prematurely (*Figure 1C,*

C'). To explore these possibilities, live imaging of LY411575 treated embryos was performed (*Figure 1H*). Compared to controls, Notch inhibition led to a significant increase in symmetric neurogenic divisions that generated two Ath5+ progenitors (48% compared to 8% in controls, *Figure 1I*, *Figure 1—video 2*). Thus, Notch signalling regulates the asymmetry of the division that produces neurogenic Ath5+ progenitors. Interference with Notch and thereby this division asymmetry influences neurogenesis patterns and tissue maturation in the developing retina.

## Ath5 positive progenitors generally translocate more basally than Ath5 negative sister cells but a spatial bias is not observed at the cell population level

The selection of the sister cell that enters the neurogenic program could be regulated by differential exposure to the extracellular environment that hosts different fate determinants as shown for other systems (*Yamashita and Fuller, 2005*; *Knoblich, 2008*). It was previously reported that Notch RNA is distributed in a graded manner along the cell's apicobasal axis with more Notch RNA detected towards apical positions in the retinal neuroepithelium (*Murciano et al., 2002*; *Del Bene et al., 2008*). This outcome led to the idea that distinct Notch signalling milieus exist along the apicobasal axis of the retinal neuroepithelium and that these could influence neurogenic outcome due to nuclear movement along this axis during IKNM (*Murciano et al., 2002*; *Baye and Link, 2007*; *Del Bene et al., 2008*). Based on these studies, we asked whether the observed cell division asymmetry that generates Ath5+ progenitors (*Figure 1C*) could be explained by differential translocation depths between Ath5+ and Ath5- sister cells. To this end, we compared trajectories of Ath5+ and Ath5- sister cells after division (*Figure 2A*). We found that indeed one sister cell soma always translocated more basally than the other (*Figure 2B*), with an average difference of 10.11 ± 7.292 µm between the maximal basal positions (n = 35 divisions, N = 9 embryos; *Figure 2C*). In 78% of sister cells, the more basally translocating sister cell started expressing Ath5 (27/35 divisions; *Figure 2D*) while in 22% of divisions the Ath5- cell positioned the soma more basally (*Figure 2E*). In the latter case, however, the difference between maximum basal position of sister cells was on average less prominent than in the reverse scenario (difference of 6.048 ± 3.559 µm when Ath5- soma is most basal compared to 11.31 ± 7.712 µm when Ath5+ soma is most basal; *Figure 2F*).

To probe whether a basal positioning threshold could be defined that predicts the onset of Ath5 expression, all trajectories were pooled (*Figure 2G,H*) To our surprise, this analysis showed a significant overlap between trajectories and maximum basal positions for Ath5+ and Ath5- somas without clearly distinct Ath5+ and Ath5- zones (*Figure 2G,H,I*, *Figure 2—figure supplement 1A*). The measurement of the average depth occupied by somas of Ath5+ and Ath5- cells further confirmed that the two populations did not occupy distinguishable basal positions (integral mean, *Figure 2—figure supplement 1B*).

These results show that basal translocation differs between sister cells and that the cell becoming Ath5+ usually translocated to more basal positions. Nevertheless, basal position was not a clear-cut readout for Ath5 expression onset as an overlap zone existed between Ath5+ and Ath5- cells.

## Asymmetric neurogenic commitment is independent of maximum basal position

The large overlap detected for basal positioning of Ath5+ and Ath5- sister cell soma argued against a direct link between basal translocation and neurogenic commitment. To show this more directly, we set out to equalise the basal translocation pattern after apical divisions of sister cells. To achieve this, we used a heat shock inducible dominant negative version of dynactin (*Norden et al., 2009*), as dynactin downregulation has been shown to induce increased basal translocation of retinal progenitor cell soma (*Del Bene et al., 2008*; *Norden et al., 2009*; *Figure 2J*). Heat shock expression of the hsp70:DNdynactin-mKate2 construct led to the translocation of both sister cell somas to greater and almost identical basal positions compared to controls (*Figure 2K*, *Figure 2—video 1*, *Figure 2—figure supplement 1E*). However, despite the fact that both sister cell somas reached very basal positions, only one became Ath5+ while the other remained Ath5- (n = 10, N = 5) (*Figure 2L*, *Figure 2—video 1*).

These results suggest that the asymmetry of neurogenic commitment is not linked to differential somal translocation of sister cells along the apicobasal axis.

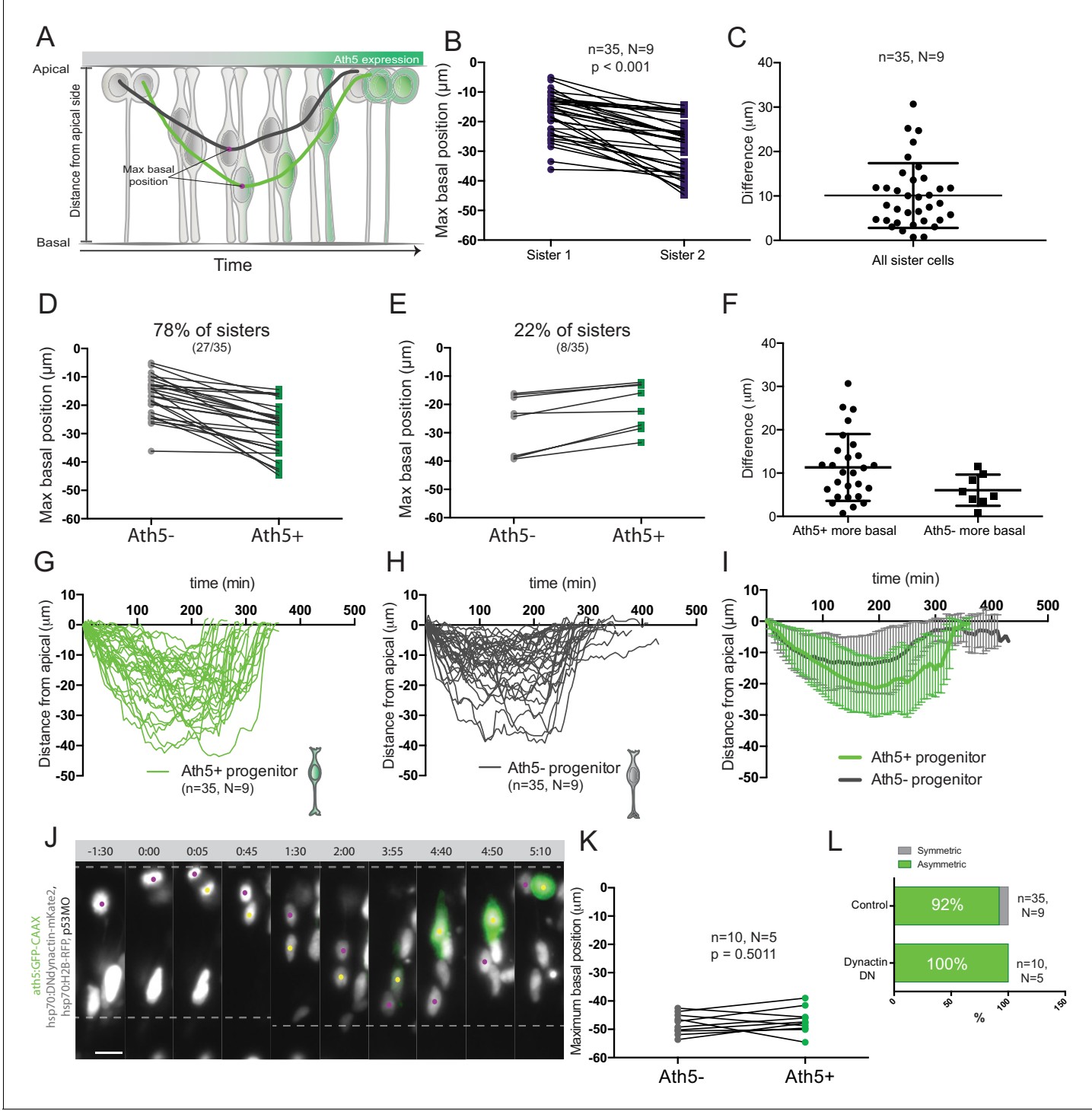

**Figure 2.** Asymmetry of division generating Ath5+ progenitors is independent of basal soma positioning in the neuroepithelium. (**A**) Schematic of progenitor cell soma moving along the apicobasal axis between divisions. (**B**) Maximum basal position of sister cells. Paired *t*-test was used to compare sister cells. Lines connect sister cells. (**C**) Difference in maximum basal position between sister cells. Vertical error bars represent standard deviation. (**D**) Maximum basal position of sister cells in cases in which Ath5+ sister cells translocate more basally (27/35, 78%). (**E**) Maximum basal position of sister cells in cases in which Ath5- sister cells translocate more basally (8/35, 22%). (**F**) Difference in maximum basal position between sister cells, comparing Ath5+ and Ath5- cells at most basal positions. (**G**) Ath5+ progenitor trajectories between divisions. Start = 0 min, mitosis of mother cell. End, onset of cell rounding. (**H**) Ath5- progenitor trajectories between divisions. Start = 0 min, mitosis of mother cell. End, onset of cell rounding. (**I**) Mean + standard deviation of Ath5+ and Ath5- progenitor pooled tracks from panels G (Ath5+, green) and H (Ath5-, grey). (**J**) Asymmetric division upon DN-dynactin overexpression. hsp70:H2B-RFP (nuclei, grey), hsp70:DNdynactin-mKate2 (dynactin, grey) ath5:GFP-CAAX (Ath5, green). Scale bar, 10 µm. Dashed lines

*Figure 2 continued on next page*

*Figure 2 continued*
show apical and basal sides of the neuroepithelium. Magenta and yellow dots label sister cells. (K) Maximum basal position of sister cells upon dynactin inhibition. Paired *t*-test to compare sister cells. (L) Percentage of asymmetric and symmetric divisions observed upon disruption of dynactin function compared to control.

The online version of this article includes the following video, source data, and figure supplement(s) for figure 2:

**Source data 1.** Source data for panels B, C, D, E, F, G, H and K. .
**Figure supplement 1.** Depth of basal translocation is not linked to sister cell fate.
**Figure supplement 1—source data 1.** Source data for panels B, C and D of *Figure 2—figure supplement 1*.
**Figure 2—video 1.** Somal translocation of sister cells after asymmetric divisions upon dynactin inhibition.
https://elifesciences.org/articles/60462#fig2video1

## Notch pathway components are uniformly distributed along the apicobasal axis of the retinal neuroepithelium

Our results so far argue against a graded effect of Notch signalling on the asymmetry of divisions producing Ath5+ progenitors. Thus, we probed whether the previously reported Notch gradient at the RNA level (*Del Bene et al., 2008*; data not shown) was also found at the level of the signalling proteins.

We performed immunostaining against two components of the Notch pathway, the receptor Notch1b and the ligand DeltaC. We found that both Notch pathway components appeared as puncta that were uniformly distributed along the apicobasal axis of the retinal neuroepithelium (*Figure 3A,A',A''* and *Figure 3B,B'*). The specificity of the Notch1b and DeltaC antibodies was confirmed by somite labelling (*Figure 3—figure supplement 1C*). Further, antibody staining in notch1b and deltaC morphant embryos showed a disappearance respectively a strong reduction of labelling along the apicobasal axis of the neuroepithelium (*Figure 3—figure supplement 1A,B*). We further analysed the distribution of Notch signalling activity using two previously established reporter lines: Tg(her4.1:mcherry-CreERT2), a marker for the gene her4, a direct Notch target (*Takke et al., 1999*), and Tg(Tp1bglob:H2BmCherry), a marker for a secondary Notch target (*Souilhol et al., 2006*). This analysis confirmed that Ath5- cells that display active Notch can occupy diverse positions along the apicobasal axis of the neuroepithelium (*Figure 3C,D* and *Figure 3—video 1*). Interestingly, Ath5+ progenitors never seemed to express Tp1 or her4 (*Figure 3C,D*, arrowheads), while Ath5- cells were seen to express both of these markers. Quantification of cells with Tp1 expression confirmed this notion, as we found that 210/215 of Ath5+ cells showed no Tp1 expression while Tp1 expression was very weak in the remaining 5 cells (n = 215 cells, N = 4 embryos; *Figure 3—figure supplement 2A,A'*). Thus, Ath5+ and Ath5- sister cells experience different levels of Notch signalling activation, independently of their position along the apicobasal axis.

Overall, these results, combined with the fact that asymmetric Ath5 expression onset persisted even when both sister cell somas translocated very basally, argue that Notch signalling does not act via an apicobasal signalling gradient for setting up the asymmetry of divisions leading to neurogenic commitment.

## Basal somal translocation of Ath5 positive progenitors depends on a combination of basal process inheritance and stabilised microtubules

While we show that somal positioning along the apicobasal axis does not influence the asymmetry of divisions producing Ath5+ progenitors, we nevertheless found differences in basal translocation potential between Ath5+ and Ath5- sister cells. In the majority of cases, the cell that later starts expressing Ath5 (Ath5+ progenitor) translocated more basally (*Figure 2D*). We thus asked what other factors could facilitate this enhanced basal translocation pattern of Ath5+ progenitors.

To this end, we compared basal translocation kinetics of Ath5+ and Ath5- progenitor soma and performed mean square displacement (MSD) analysis. This analysis revealed that Ath5+ basal somal translocation is more directed than that of Ath5- soma (*Figure 4B*), a finding confirmed by directionality ratio analysis (*Figure 2—figure supplement 1C*). Furthermore, Ath5+ sister cell soma showed a higher average velocity during basal translocation (*Figure 2—figure supplement 1D*), indicating that Ath5+ progenitors translocate more efficiently towards the basal side.

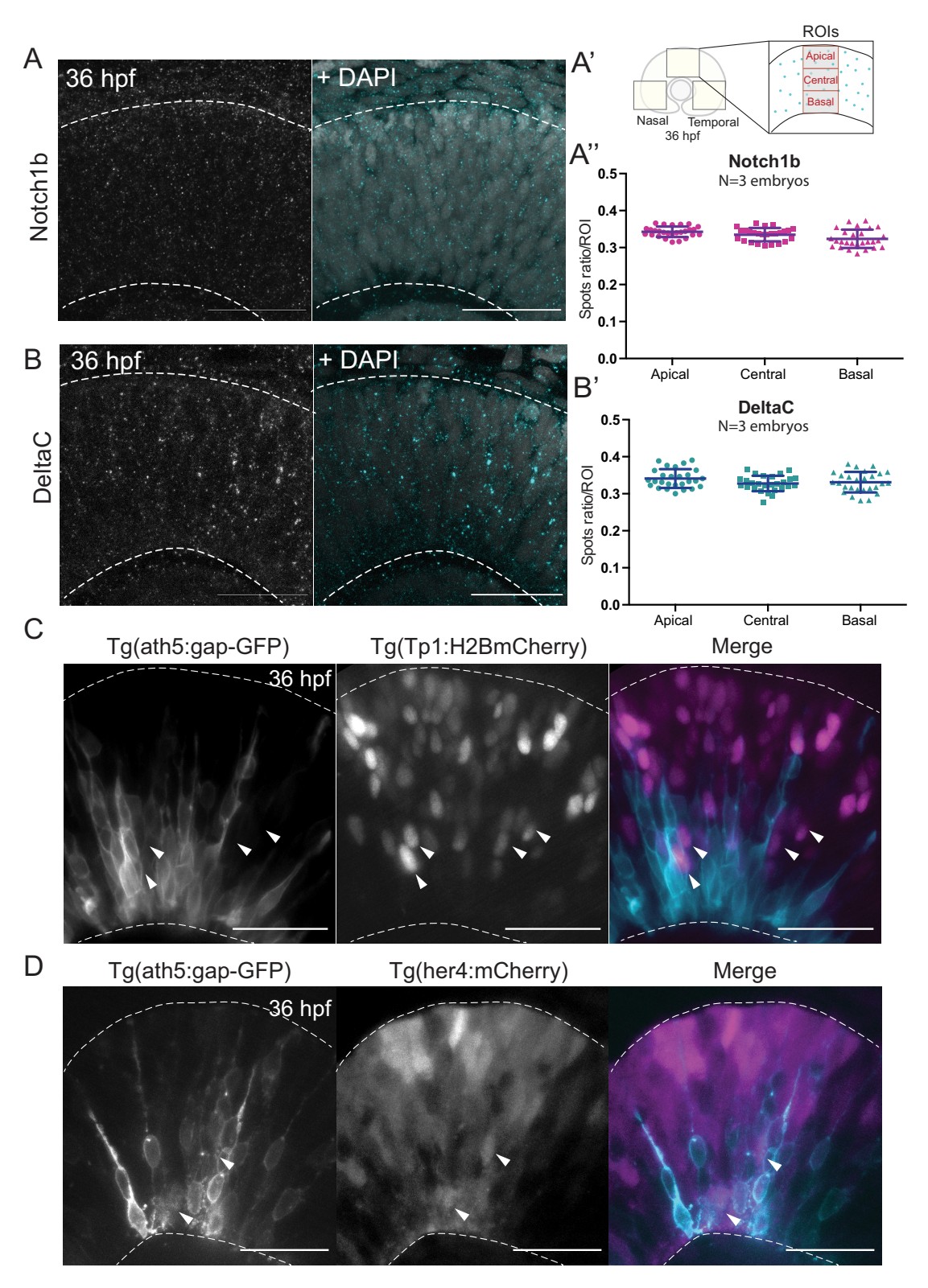

**Figure 3.** Notch signalling components are uniformly distributed along the apicobasal axis of the retinal neuroepithelium. (**A**) Staining 36 hpf embryos for Notch1b. Dashed lines show apical and basal sides of the retinal neuroepithelium. Scale bar, 30 μm. DAPI labels nuclei. (**A'**) Schematic of regions of interest (ROIs) along the apicobasal axis used to measure the DeltaC and Notch1b spot ratio. (**A''**) Spot ratio distribution for Notch1b along the apicobasal axis. Unpaired *t*-test with Welch's correction. P values: central vs apical, p=0.0896; basal vs central, p=0.0630; basal vs apical, p=0.012. (**B**)
*Figure 3 continued on next page*

*Figure 3 continued*

Staining of 36 hpf embryos for DeltaC. Dashed lines show apical and basal sides of the retinal neuroepithelium. Scale bar, 30 µm. DAPI labels nuclei. (**B'**) Spot ratio distribution for DeltaC along the apicobasal axis. Unpaired *t*-test with Welch's correction. P values: central vs apical, p=0.0759; basal vs central, p=0.6697; basal vs apical, p=0.3053. (**C**) Images of retinae expressing Tp1:H2BmCherry (magenta) and ath5:gap-GFP (cyan) at 36 hpf. Scale bar, 30 µm. Arrows show Tp1 positive Ath5 negative cells. Dashed lines show apical and basal sides of the retinal neuroepithelium. (**D**) Images of retinae expressing her4:mCherry (magenta) and ath5:gap-GFP (cyan) at 36 hpf. Scale bar, 30 µm. Arrows show her4 positive but Ath5 negative cells. Dashed lines show apical and basal sides of the retinal neuroepithelium.

The online version of this article includes the following video, source data, and figure supplement(s) for figure 3:

**Source data 1.** Source data for panels A'' and B'.
**Figure supplement 1.** Controls for Notch1b and DeltaC stainings in the retinal neuroepithelium.
**Figure supplement 2.** Ath5+ progenitors do not carry active Notch signalling.
**Figure 3—video 1.** Progenitors showing active Notch signalling occupy the entire apicobasal axis.
https://elifesciences.org/articles/60462#fig3video1

Basal process inheritance is known to facilitate efficient basal translocation of progenitors and emerging neurons in neocortex and retinal neuroepithelia (*Saito et al., 2003*; *Icha et al., 2016a*). We thus asked whether basal process inheritance was correlated with faster basal displacement of Ath5+ soma (*Figure 4A*). Indeed, in 15 out of 16 cell pairs analysed, the cell that inherited the basal process initially translocated its soma more basally (*Figure 4C*). However, only in 10/16 cases (60%), did that cell also become the Ath5+ progenitor (*Figure 4D*). Thus, it was possible that additional factors were responsible for the more efficient basal somal translocation of Ath5+ cells. It was previously shown that stabilised microtubules emanating from the apically positioned centrosome facilitate somal translocation of emerging retinal ganglion cells, the first neurons born in the retina (*Icha et al., 2016a*). To test whether stabilised microtubules could also drive translocation of Ath5+ progenitors, we stained for acetylated tubulin in Tg(ath5:gap-GFP/RFP) embryos (*Zolessi et al., 2006*) at stages before and during Ath5+ progenitors emergence, 24, 28, 32, and 36 hpf (*Figure 4E*). This revealed that at 24 hpf, the developmental stage prior to neurogenesis onset, acetylated tubulin was seen only in the primary cilium of progenitor cells (*Figure 4E*, top left). However, once neurogenesis started, acetylated tubulin was also observed in the apical process of progenitor cells (*Figure 4E* top right and bottom left, 28 and 32 hpf) as well as in retinal ganglion cells at 36 hpf (*Figure 4E* bottom right), as previously reported (*Icha et al., 2016a*).

To examine whether these stabilised microtubules were directly involved in the basal somal translocation of neurogenic progenitors, we backtracked Ath5+ progenitors overexpressing the microtubule-destabilising protein Stathmin-1 using an hsp70:Stathmin1-mKate2 construct (*Figure 4F*, *Figure 4—video 1*; *Icha et al., 2016a*). The soma of Ath5+ progenitors expressing Stathmin-1 did not translocate as basally as the soma of control cells (*Figure 4I,K*). In contrast, the basal somal translocation of Ath5- progenitors was not affected (*Figure 4H,J*), indicating that only progenitors that enter the neurogenic state use stabilised microtubules for basal somal translocation. This experiment further confirmed that Ath5 expression onset is independent of somal positioning along the apicobasal axis. While both sister cells remained at very apical positions after division, only one became an Ath5+ progenitor (*Figure 4G*, 10/10 divisions). This is the opposite scenario to the dominant negative dynactin condition in which cell nuclei translocated more basally (*Figure 4J,K*). Thus, the asymmetry of Ath5 expression onset persisted independently of sister cell basal or apical positions (*Figure 2L*, *Figure 2—video 1*, *Figure 4—video 1*).

Overall, these experiments showed that the greater basal somal translocation of Ath5+ progenitors is driven by a combination of basal process inheritance and the presence of apically enriched stabilised microtubules. This reflects other examples of neuronal basal somal translocation in CNS development (*Yuasa et al., 1996*; *Umeshima et al., 2007*; *Cooper, 2013*; *Sakakibara et al., 2014*; *Icha et al., 2016a*).

## Notch signalling affects neurogenic commitment through Sara endosomes

We conclude that the greater basal translocation of progenitors that start expressing Ath5 was driven by a combination of basal process inheritance and a stabilised microtubule network. However, these findings do not yet explain the involvement of Notch signalling in the asymmetry of

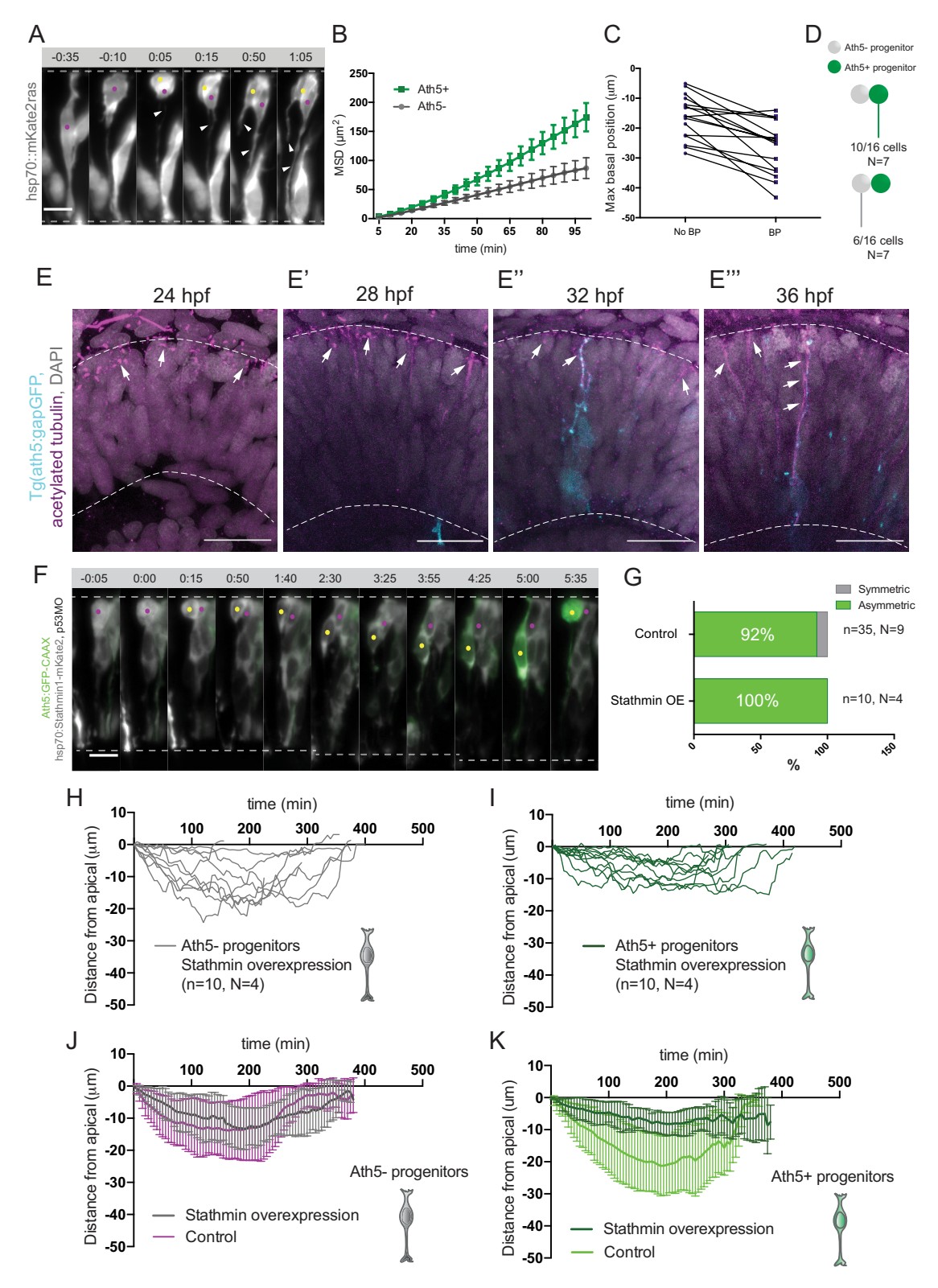

**Figure 4.** Basal soma translocation of Ath5+ progenitors is facilitated by basal process inheritance and stabilised microtubules. (**A**) Asymmetric inheritance of basal process during progenitor division. hsp70:mKate2-ras labels cell membrane (grey). Arrowheads: basal process. Scale bar, 10 μm. Magenta and yellow dots label sister cells. Dashed lines show apical and basal side of retinal neuroepithelium. (**B**) Mean square displacement (MSD) for Ath5+ and Ath5- progenitors, calculated for basal translocation within the first 100 min after mitosis of the mother cell. Data from *Figure 2G and H*. (**C**)

*Figure 4 continued*

Maximum basal position of sister cells not inheriting the basal process (No BP) vs inheriting the basal process (BP) after division. (D) Proportion of Ath5+ and Ath5- progenitor cells inheriting the basal process. (E) Acetylated tubulin staining (magenta) of Tg(ath5:gap-GFP, cyan) with nuclei labelled by DAPI (grey) at 24 (upper left panel), 28 (upper right), 32 (bottom left) and 36 hpf (bottom right). Scale bar, 30 µm. Arrows mark acetylated tubulin staining in the primary cilium (upper left), retinal progenitors (upper right and bottom left) and retinal ganglion cells (bottom right). Dashed lines show apical and basal sides of the retinal neuroepithelium. (F) Example of basal translocation of progenitors upon Stathmin1 overexpression induced at 28 hpf. hsp70:Stathmin1-mKate2 (stathmin, grey), ath5:GFP-CAAX (Ath5, green). Scale bar, 5 µm. Magenta and yellow dots label sister cells. Dashed lines show apical and basal sides of the retinal neuroepithelium. (G) Percentage of asymmetric and symmetric divisions observed upon Stathmin1 overexpression compared to control. (H) Ath5- progenitor trajectories between divisions upon Stathmin1 overexpression. Start = 0 min, mitosis of mother cell. End, onset of cell rounding. (I) Ath5+ progenitor trajectories between divisions upon Stathmin1 overexpression. Start = 0 min, mitosis of mother cell. End, onset of cell rounding. (J) Mean and Standard Deviation of Ath5- progenitor trajectories in control (magenta) and Stathmin1 overexpression (grey). (K) Mean and Standard Deviation of Ath5+ progenitor trajectories in control (light green) and Stathmin1 overexpression (dark green).

The online version of this article includes the following video and source data for figure 4:

**Source data 1.** Source data for panels B, C, H and I.
**Figure 4—video 1.** Somal translocation of sister cells after asymmetric divisions upon stathmin overexpression.
https://elifesciences.org/articles/60462#fig4video1

---

neurogenic commitment independently of a Notch activity gradient. It was shown that Notch signalling components can segregate asymmetrically during division of the mother cell in other systems (*Le Borgne and Schweisguth, 2003*; *Dong et al., 2012*; *Kechad et al., 2012*; *Coumailleau et al., 2009*), which leads to the inheritance of different levels of Notch by the two daughter cells. In particular, Sara-positive endosomes (Sara endosomes) that carry active Notch via the endocytosis of Delta receptors and Notch ligands were shown to be involved in asymmetric neurogenic fate decisions in the *Drosophila* sensory organ precursor and the zebrafish spinal cord (*Coumailleau et al., 2009*; *Kressmann et al., 2015*).

To test whether Sara endosomes could also be involved in divisions generating Ath5 positive progenitors, we assessed whether Sara- endosomes are present in retinal progenitors at stages at which Ath5+ progenitors emerge. Embryos were injected with mRFP-Sara mRNA and stained for Notch1b and DeltaC. We found that these components indeed co-localise in retinal progenitors (*Figure 5A, A'* and *Figure 5B,B'*). To understand whether Sara endosomes are asymmetrically distributed between progenitors upon division as suggested elsewhere (*Coumailleau et al., 2009*; *Kressmann et al., 2015*), we injected mRFP-Sara RNA into membrane labelled Tg(actb1:HRAS-EGFP) embryos. At all developmental stages analysed, 24, 28 and 36 hpf, asymmetric distribution of Sara endosomes was observed in dividing cells (*Figure 5C*). Cells mainly contained 1–3 endosomes and rarely more than that (*Figure 5D*, n = 25 dividing cells, N = 7 embryos). In most cases, one of the two sister cells inherited all endosomes (*Figure 5D'*), suggesting that the majority of Sara endosome related Notch signalling is dispatched to only one sister cell. We further confirmed that Tp1-expressing cells carry Sara endosomes (*Figure 3—figure supplement 2B*) meaning that they feature active Notch signalling through the endosomal pathway at 36 hpf. Furthermore, light-sheet live imaging confirmed asymmetric distribution of Sara- endosomes during multipotent retinal progenitor divisions (*Figure 5E,E'*). These experiments indicated that Notch signalling is asymmetrically partitioned through Sara-positive endosomes. Thus, Sara-endosomes inheritance could mediate asymmetric neurogenic commitment in the retina. Our finding that Ath5+ progenitors never displayed active Notch signalling (*Figure 3C,D*, *Figure 3—figure supplement 2A*) is in agreement with this notion. Combined, these findings argue that the depletion of Sara endosomes is linked to Ath5+ fate. We propose that Sara endosomes are mostly inherited by the Ath5- cell, which retains high Notch signalling and thus stays in the proliferative state (see working model in *Figure 5G* and cell in *Figure 5—figure supplement 1E*, *Figure 5—video 1*).

To further test the involvement of Sara endosomes in asymmetric neurogenic commitment, we depleted the Sara protein using an established Sara splice morpholino (*Kressmann et al., 2015*). We found that the number of Ath5+ cells was greatly reduced compared to controls upon Sara inhibition, leading to the presence of fewer neurons (*Figure 5F*, *Figure 5—figure supplement 1A*). At the same time, the number of Tp1-positive cells increased (*Figure 5F'*) suggesting an increase in active Notch signalling in both daughter cells. This is the opposite scenario of what is observed

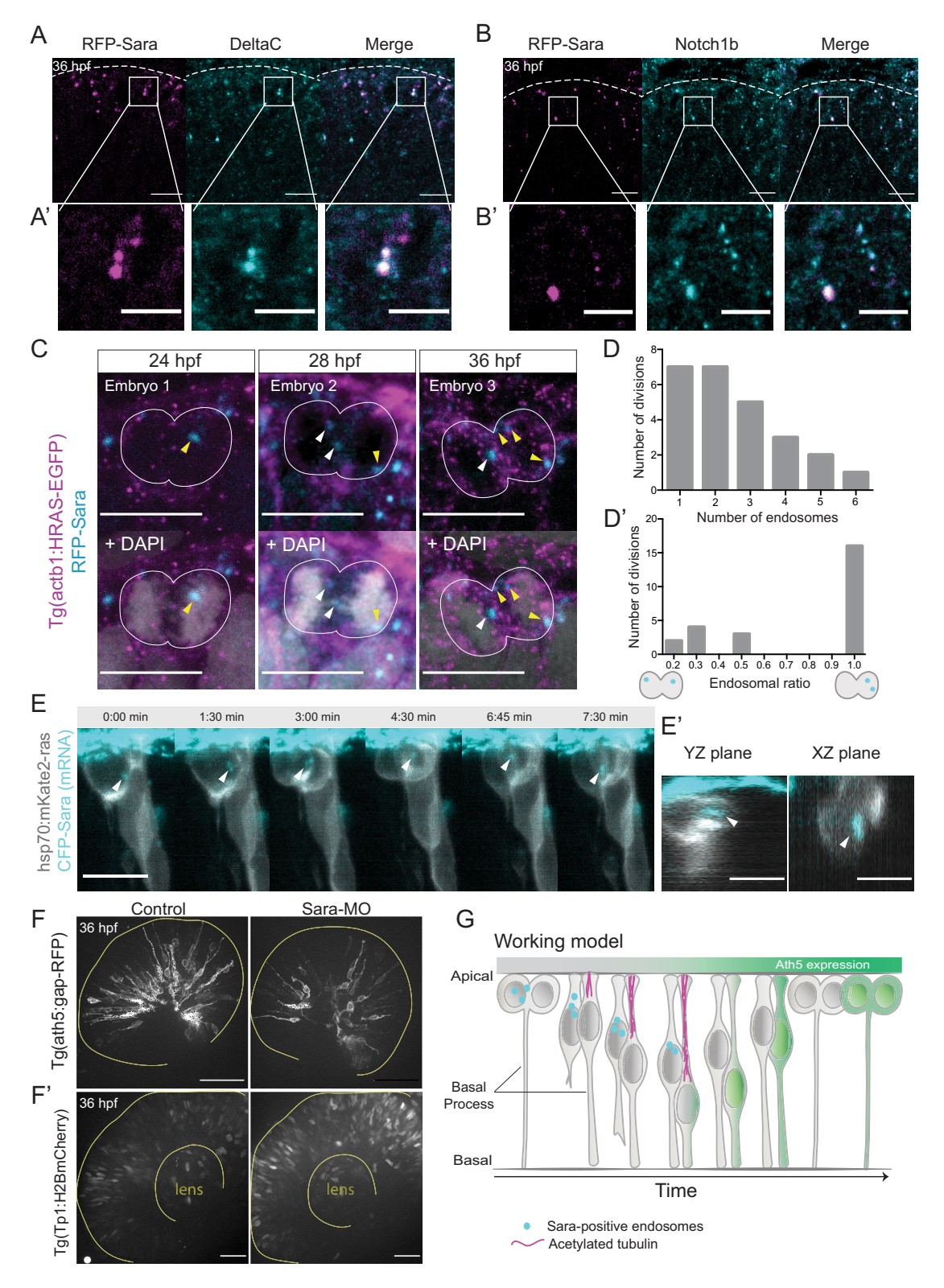

**Figure 5.** Notch signalling affects neurogenic commitment through Sara endosomes. (**A**) Co-localisation of Sara-positive endosomes (RFP-Sara, magenta) and DeltaC staining (cyan) in 36 hpf embryos. Scale bar, 10 μm. Dashed line represents the apical side. (**A'**) Close-up of co-localisation. Scale bar, 5 μm. (**B**) Co-localisation of Sara-positive endosomes (RFP-Sara, magenta) and Notch1b staining (cyan) in 36 hpf embryos. Scale bar, 10 μm. Dashed line represents the apical side. (**B'**) Close-up of co-localisation. Scale bar, 5 μm. (**C**) Asymmetric distribution of Sara-positive endosomes in

*Figure 5 continued on next page*

*Figure 5 continued*

dividing progenitor cells in three different embryos fixed at three different developmental stages, 24, 28 and 36 hpf. Tg(actb1:HRAS-EGFP) (membrane, magenta), RFP-Sara (Sara-positive endosomes, cyan), DAPI (chromatin, grey). Scale bar, 10 µm. (D) Number of endosomes within dividing cells. n = 25 cells, 7 embryos. (D') Histogram of endosomal ratio between dividing cells. n = 25 cells, 7 embryos. (E) Asymmetric distribution of Sara-positive endosomes in dividing cells in live samples at 32 hpf. hsp70:mkate2-ras (membrane [grey], CFP-Sara [Sara-positive endosomes, cyan]). Scale bar, 10 µm. Arrowheads point to the endsosomes (E') YZ (left) and XZ (right) view of sister cells (grey) and Sara endosomes (cyan) at minute 3:00. Scale bar, 10 µm. Arrowheads point to the endsosomes. (F) Ath5+ cells (grey) in retinal neuroepithelium at 36 hpf in control (left) and Sara knockdown (Sara-MO, right). Scale bar, 50 µm. The yellow line delimits the apical side of the retinal neuroepithelium. (F') Tp1+ cells (grey) at 36 hpf in control (left) and Sara morpholino knockdown (right). Scale bar, 30 µm. The yellow line delimits the apical side of the retinal neuroepithelium and the lens. (G) Scheme recapitulating the main findings of this study presenting a working model for Sara-positive endosomes inheritance and its role in asymmetric divisions. The online version of this article includes the following video and figure supplement(s) for figure 5:

**Figure supplement 1.** Sara-positive endosomes are asymmetrically distributed during cell division.

**Figure 5—video 1.** Asymmetric inheritance of Sara endosomes during asymmetric progenitor division.

https://elifesciences.org/articles/60462#fig5video1

upon Notch signalling inhibition which led to the generation of more neurogenic progenitors through an increase in symmetric neurogenic divisions (*Figure 1E,H,I*). We excluded the possibility that this decrease in Ath5+ progenitor emergence is due to cells not being able to divide and progress through the cell cycle before neurogenesis, as pH3 staining showed similar amounts of mitotic cells in Sara morphants as in control embryos (*Figure 5—figure supplement 1B*). Furthermore, Sara morphants showed a higher number of EdU-positive cells than embryos injected with control morpholino, suggesting that indeed more cells remained proliferative (*Figure 5—figure supplement 1C,D*).

We thus speculate that depletion of Sara led to more symmetric proliferative divisions generating two Ath5- cells. This finding differs from an earlier study in which Sara depletion led to an increase of neuronal production (*Kressmann et al., 2015*). However, in this study a different neuronal lineage was investigated, raising the possibility that diverse mechanisms of Sara-mediated regulation of Notch exist that are dependent on tissue and lineage context (see discussion). Overall, these results suggest that Sara-mediated intracellular Notch signalling is necessary to create asymmetries in Notch activity among progenitors, thereby selecting the progenitors that enter the neurogenic path.

## Discussion

We here used long-term live imaging to explore the emergence of neurogenic progenitors in the vertebrate retina. Our study reveals that early neurogenic commitment arises via asymmetric divisions of multipotent retinal progenitors that generate one Ath5+ neurogenic progenitor and one Ath5- progenitor that will enter a different lineage. This asymmetry is regulated by Notch signalling and is important to ensure the balance between proliferating cells and cells entering differentiation routes at early neurogenesis stages. Interference with the division asymmetry leads to premature or delayed differentiation. We further provide evidence that Notch does not act through an apicobasal activity gradient as previously proposed but via the endocytic pathway. *Figure 5G* summarises our main findings and presents a working model.

Our quantitative imaging approach showed that progenitor divisions that generate the earliest retinal neurogenic progenitors, Ath5 positive progenitors, are reproducibly asymmetric. Notch inhibition leads to more symmetric neurogenic divisions producing two neurogenic progenitors. This explains the increase in the total number of neurons at the lineage level, adding to previous findings that investigated this issue at the tissue level (*Austin et al., 1995*; *Ahmad et al., 1997*). This increase in symmetric divisions eventually leads to the premature exhaustion of multipotent progenitors.

In line with our findings, it has been shown earlier that Notch is involved in asymmetric divisions that generate two different neurons or one neuron and one proliferative cell in other neural systems. including the vertebrate spinal cord and mammalian neocortex (*Mizutani et al., 2007*; *Nelson et al., 2013*; *Kressmann et al., 2015*). For the early neurogenic progenitor divisions in the retina investigated here, it was speculated that Notch acts via a signalling gradient along the apicobasal axis of the neuroepithelium (*Murciano et al., 2002*; *Del Bene et al., 2008*). It was further proposed that more basally translocating progenitors are more likely to become neurogenic, generating two

neurons after the next division (*Murciano et al., 2002*; *Baye and Link, 2007*). This led to the hypothesis that Notch signalling acts mainly at the apical side of the neuroepithelium where it suppresses neurogenic fate. However, due to technical challenges at the time, these analyses could only take a small number of unrelated cells into consideration. It was reported in one of these previous studies that tissue wide treatment of chick retinae with cytochalasin B led to increased premature neurogenesis (*Murciano et al., 2002*). However, no tracking of nuclei was performed and the tissue wide effects that such drug treatment entail, including cell delamination, were not further investigated. We thus aimed to manipulate nuclear position affecting single cells, to have the possibility to track the basal translocation patterns of large cohorts of sister cells. This generated a more quantitative and direct assessment of the effects of nuclear positioning during IKNM on neurogenesis in the context of the first asymmetric neurogenic divisions. We indeed found that the more basally translocating sister cell is more likely to enter the Ath5+ lineage. However, when the entirety of Ath5+ cells was taken into account, a significant overlap of basal positioning of Ath5+ and Ath5- cohorts was observed. In addition, the asymmetry of neurogenic commitment persisted when the translocation position of sister cells was levelled towards either the basal or the apical side. Together, this shows that somal position along the apicobasal axis of the retinal neuroepithelium does not influence asymmetric neurogenic commitment.

Instead, we present evidence that it is the asymmetric partitioning of Sara endosomes that drives the asymmetry of retinal progenitor divisions to enter the neurogenic pathway. A similar idea was previously proposed for the *Drosophila* sensory organ precursor and the zebrafish spinal cord (*Coumailleau et al., 2009*; *Kressmann et al., 2015*). Ideally, we would have directly correlated Sara-positive endosomes inheritance with neurogenic or proliferative fate occurrence in numerous cells. However, this correlation could not be systematically quantified over several instances of asymmetric divisions in our system, due to its complexity and long timeframes between division and fate read-out. Nevertheless, we could follow one division in which Sara endosomes were asymmetrically inherited and assess the fate of the sister cells. As expected, the cell inheriting all endosomes became the Ath5- progenitor and thus remained proliferative (*Figure 5—figure supplement 1E*, *Figure 5—video 1*), while the cell that did not inherit Sara endosomes started Ath5 expression and thereby entered the neurogenic lineage (see working model *Figure 5G* and *Figure 5—figure supplement 1E*, *Figure 5—video 1*). This outcome is in line with the finding that Ath5+ cells do not show active Notch signalling. These observations, combined with the finding that asymmetric inheritance is observed at stages at which asymmetric Ath5 decisions occur and that Sara knockdown impairs the balance between proliferation and neurogenesis in the Ath5 lineage, strongly argue that Notch signalling via Sara-positive endosomes is important for lineage asymmetry in the retina. The asymmetric distribution of Sara-positive endosomes most likely leads to the downregulation of Notch in one sister cell. When Sara is depleted however, more progenitors seem to express active Notch signalling, leading to a reduced amount of neurogenic Ath5+ cells at early neurogenic stages. This finding is different from what was presented in an earlier study that investigated the role of Sara endosomes on neurogenesis in the zebrafish spinal cord (*Kressmann et al., 2015*). Here, Sara depletion (mutant and morphant) had the same effect as direct inhibition of Notch signalling, namely the premature production of neurons. This argues that Sara can fine-tune Notch activity differently in different ttissues and neuronal lineage contexts. We speculate that in the context of the early retinal lineages we studied, asymmetric neurogenic decisions are influenced and regulated by Sara protein that guides the asymmetric dispatch of these endosomes (*Coumailleau et al., 2009*), consequently downregulating Notch in one of the sister cells and promoting Notch in the other. When Sara is not present however, Notch continues to be active. In this case, Notch distribution between sister cells is not regulated any more through the endosomal pathway, so it cannot be depleted from one sister cell during division, and as a consequence more cells maintain high Notch activity. These additional cells then also stay in the proliferative state, leading to a decrease in the number of neurogenic cells. The exact mechanism of how Sara fine-tunes lineage decisions in different contexts will be interesting to explore in future studies. One factor could be the commitment state of neurogenic progenitors which differs between spinal cord and retina, but this point will need further investigation.

It should be noted that our systematic analysis not only enhances the understanding of Notch signalling in this particular neurogenesis event, but also shows the importance of quantitative approaches in developmental systems. Only due to the analysis of large cohorts of sister cells combined with a backtracking approach could we achieve the depth of analysis needed to exclude the

role of a possible Notch gradient, which led to the investigation of the Notch endocytic pathway. The advance of imaging techniques, image analysis algorithms, and the possibility of performing backtracking studies in the retina are exciting emerging tools to follow similar questions in future studies. Such sophisticated studies will in turn promote our understanding of reproducibility and/or plasticity of lineage decisions at diverse developmental stages in different tissues of the CNS (*Zechner et al., 2020*).

Despite the fact that basal position is not directly linked to Ath5+ fate commitment, we find that the majority of progenitors entering the Ath5 lineage nevertheless translocate their somas more basally than their Ath5- sister cell. This observation argues that the more basal translocation of neurogenic Ath5+ progenitors is a consequence of neurogenic commitment, rather than a cause for it. We find that basal translocation of Ath5+ progenitors is linked to apically localised stabilised microtubules and basal process inheritance. This is remarkably similar to what has been shown for somal translocation events of retinal ganglion cells (*Icha et al., 2016a*). By contrast, this is different to nuclear and thereby somal translocation during IKNM in multipotent progenitors, which has been shown to be caused by passive nuclear displacements (*Norden et al., 2009*; *Kosodo et al., 2011*; *Leung et al., 2011*). Thus, an interesting interpretation of these findings could be that Ath5+ progenitors constitute an intermediate cell type placed between multipotent progenitors and differentiating neurons at the level of cellular arrangements including cytoskeletal components. This is reminiscent of intermediate progenitors found in the mammalian neocortex (*Haubensak et al., 2004*; *Miyata et al., 2004*; *Noctor et al., 2004*). To our knowledge, the existence of such intermediate progenitors has not been thoroughly investigated in the retina. However, a recent study performing single cell RNA sequencing of mouse retinae (*Wu et al., 2020*) hints that a transitional state of retinal progenitors that express atoh7 and other neurogenic genes could exist. Therefore, it will be interesting to probe how cytoskeletal changes could be linked to these cell identity changes in the developing retina and beyond. Such studies will lead to new insights into the biology of fate decisions downstream of transcription profiles and towards the cell biological scale.

Asymmetric progenitor divisions are an important mean to maintain the balance between proliferation and differentiation in developing organs. To understand the regulation of these types of divisions and thereby embryogenesis as a whole, their investigation needs to span scales from the molecular to the cellular to the tissue level. Furthermore, investigations need to combine knowledge of cell type specific transcription factor profiles, the signalling factors involved in guiding fate decisions and the cell biological consequences that differentiation entails. The emerging possibilities of quantitative in vivo studies mentioned above in combination with the advent of single-cell sequencing data that is becoming more and more extensive (*Wagner and Klein, 2020*; *Xu et al., 2020*), opens new perspectives for an understanding of these phenomena at a previously unimaginable level. This, in combination with the continued use of the zebrafish as a model for quantitative developmental biology, has the potential to reveal general principles that can then be extended to other, often less accessible systems.

# Materials and methods

## Key resources table

| Reagent type (species) or resource | Designation | Source or reference | Identifiers | Additional information |
|---|---|---|---|---|
| Genetic reagent (*Danio rerio*) | Tg(ath5:gap-GFP) | *Zolessi et al., 2006* | ID_source: ZDB-TGCONSTRCT-070129–1 | |
| Genetic reagent (*Danio rerio*) | Tg(ath5:gap-RFP) | *Zolessi et al., 2006* | ID_source: ZDB-TGCONSTRCT-070129–2 | |
| Genetic reagent (*Danio rerio*) | Tg(her4.1:mcherry-CreERT2) | *Kroehne et al., 2011* | ID_source: ZDB-TGCONSTRCT-111102–1 | |
| Genetic reagent (*Danio rerio*) | Tg(her4:EGFP) | *Yeo et al., 2007* | ID_source: ZDB-TGCONSTRCT-070612–3 | |
| Genetic reagent (*Danio rerio*) | Tg(Tp1bglob:H2BmCherry) | *Ninov et al., 2012* | ID_source: ZDB-TGCONSTRCT-120419–6 | |

*Continued on next page*

*Continued*

| Reagent type (species) or resource | Designation | Source or reference | Identifiers | Additional information |
|---|---|---|---|---|
| Genetic reagent (*Danio rerio*) | Tg(actb1:HRAS-EGFP) | *Cooper et al., 2005* | ID_source: ZDB-TGCONSTRCT-120412–2 | |
| Genetic reagent (*Danio rerio*) | Tg(hsp70:H2B-RFP) | *Dzafic et al., 2015* | | |
| Antibody | Notch1b (Zebrafish, polyclonal) | GeneTEx | RRID:AB_10625877 | IF(1:100) |
| Antibody | DeltaC (goat, monoclonal) | Abcam | RRID:AB_2043260 | IF(1:100) |
| Antibody | GFP (rabbit, polyclonal) | Proteintech | RRID:AB_11042881 | IF(1:100) |
| Antibody | Acetylated tubulin (mouse, monoclonal) | Sigma-Aldrich | RRID:AB_477585 | IF(1:250) |
| Antibody | Phospho-histone H3 (rat, monoclonal) | Abcam | RRID:AB_2295065 | IF(1:500) |
| Antibody | DAPI | Thermofisher | RRID:AB_2629482 | IF(1:1000) |
| Recombinant DNA reagent | ath5:GFP-CAAX (plasmid DNA) | *Kwan et al., 2007; Icha et al., 2016a* | RRID:Addgene_105958 | 15 ng/µl |
| Recombinant DNA reagent | hsp70:H2B-RFP (plasmid DNA) | *Strzyz et al., 2015* | RRID:Addgene_105953 | 10 ng/µl |
| Recombinant DNA reagent | hsp70:mKate2-ras (plasmid DNA) | *Strzyz et al., 2015* | RRID:Addgene_105945 | 15 ng/µl |
| Recombinant DNA reagent | hsp70:Stathmin1-mKate2 (plasmid DNA) | *Taverna et al., 2016* | RRID:Addgene_105969 | 15 ng/µl |
| Recombinant DNA reagent | hsp70:mKate2-dynactin (1-811) (plasmid DNA) | *Taverna et al., 2016* | RRID:Addgene_105970 | 15 ng/µl |
| Sequence-based reagent | Sara morpholino | *Kressmann et al., 2015* | N/A | TGAACTAGAGAC TTTACCTTGCCAC |
| Sequence-based reagent | p53 morpholino | *Robu et al., 2007* | ZFIN ID: ZDB-MRPHLNO-070126–7 | GCGCCATTGC TTTGCAAGAATTG |
| Sequence-based reagent | Standard control morpholino | Gene tools | N/A | CCTCTTACCTCAG TTACAATTTATA |
| Sequence-based reagent | notch1b morpholino | This study | N/A | GTATTGCATTCTCC TCTCCCGTCTG |
| Sequence-based reagent | deltaC-MO1 | *Zhang et al., 2008* | ZFIN ID: ZDB-MRPHLNO-050531–1 | AGCCATCTTTGCC TTCTTGTCTGCT |
| Sequence-based reagent | deltaC-MO2 | *Holley et al., 2002; Okigawa et al., 2014* | ZFIN ID: ZDB-MRPHLNO-050531–2 | CGATAGCAGACTG TGAGAGTAGTCC |
| Commercial assay or kit | EdU Click-iT-Alexa 488 | Invitrogen | ID_source:C10337 | |
| Commercial assay or kit | EdU Click-iT-Alexa 647 | Invitrogen | ID_source:C10340 | |
| Commercial assay or kit | mMessage mMachine | Ambion | ID_source:AM1344 | |
| Chemical compound, drug | DMSO | Sigma-Aldrich | ID_source:M81802 | |
| Chemical compound, drug | DAPT | Sigma-Aldrich | ID_source:D5942 | |
| Chemical compound, drug | LY411575 | Sigma-Aldrich | ID_source: T5648 | |
| Chemical compound, drug | Agarose (low gelling temperature) | Sigma-Aldrich | ID_source: A9414-250G | |
| Chemical compound, drug | MS-222 | Sigma-Aldrich | N/A | |
| Chemical compound, drug | Triton | Sigma-Aldrich | ID_source:T9284 | |
| Chemical compound, drug | PTU (N-phenylthiourea) | Sigma-Aldrich | ID_source:P7629 | |
| Software, algorithm | ZEN 2014 (black edition) | Carl Zeiss Microscopy | | |
| Software, algorithm | Andor iQ 3.6 | Andor | | |
| Software, algorithm | ZEN 2011 (black edition) | Carl Zeiss Microscopy | | |

*Continued on next page*

*Continued*

| Reagent type (species) or resource | Designation | Source or reference | Identifiers | Additional information |
|---|---|---|---|---|
| Software, algorithm | Fiji | *Schindelin et al., 2012* | | |
| Software, algorithm | Instantaneous velocities (Python 3) | This study | https://git.mpi-cbg.de/ nerli/nerli-et-al-2020-scripts | |
| Software, algorithm | Integral mean (Python 3) | This study | https://git.mpi-cbg.de/nerli/ nerli-et-al-2020-scripts | |
| Software, algorithm | Spot counter plugin (Fiji) | Nico Stuurman, ValelabUtils package | https://imagej.net/ SpotCounter | |
| Software, algorithm | ClearVolume (Fiji) | *Royer et al., 2015* | | |
| Other | CFP-Sara (mRNA) | *Kressmann et al., 2015* | | 100 pg/embryo |
| Other | mRFP-Sara (mRNA) | *Kressmann et al., 2015* | | 100 pg/embryo |

## Zebrafish husbandry

Wild-type zebrafish were maintained at 26°C. Embryos were raised at 21, 28.5, or 32°C in E3 medium, which was changed daily. To prevent pigmentation, E3 medium was supplemented with 0.2 mM 1-phenyl-2-thiourea (Sigma-Aldrich) from 8 hpf. Animals were staged in hpf according to *Kimmel et al., 1995*. All animal work was performed in accordance with European Union directive 2010/63/EU, as well as the German Animal Welfare Act.

## Transgenic lines

To visualise Ath5 neurogenic progenitors, the Tg(ath5:gap-GFP) and Tg(ath5:gap-RFP) (*Zolessi et al., 2006*) lines were used. To visualise Notch activity, the Notch reporter lines Tg(her4.1: mcherry-CreERT2) (*Kroehne et al., 2011*), Tg(her4:EGFP) (*Yeo et al., 2007*), and Tg(Tp1bglob: H2BmCherry) (*Ninov et al., 2012*) were used. To label cell membranes, the Tg(actb1:HRAS-EGFP) (*Cooper et al., 2005*) line was used. To visualise progenitors nuclei, Tg(hsp70:H2B-RFP) line was used (*Dzafic et al., 2015*).

## DNA and RNA injections

To mosaically label cells in the zebrafish retina, DNA constructs were injected into one cell stage embryos. Injected volumes ranged from 0.5 to 1 nl. The amount of injected DNA ranged from 10 to 20 pg per embryo and did not exceed 40 pg even when multiple constructs were injected. See *Table 1* for details. pCS2 expression vectors containing CFP-Sara or mRFP-Sara were kindly provided by Marcos Gonzáles-Gaitán and published previously (*Kressmann et al., 2015*). The Sara mRNA was synthesised using the Ambion mMessage mMachine kit. To visualise SARA endosomes, 100 pg of CFP-SARA or mRFP-SARA where injected into one cell stage embryos for ubiquitous expression.

## Heat shock of embryos

To induce expression of the heat shock promoter (hsp70)-driven constructs, embryos were incubated at 24 or 28 hpf in water bath at different temperatures (See *Table 1* for details). Imaging always started 3-4 hr after heat shock.

**Table 1.** DNA constructs used in this study.

| Plasmid | Concentration | Heat shock | Reference(s) |
|---|---|---|---|
| ath5:GFP-CAAX | 15 ng/µl | – | *Kwan et al., 2007*; *Icha et al., 2016a* |
| hsp70:H2B-RFP | 10 ng/µl | 37°C, 15 min | *Strzyz et al., 2015* |
| hsp70:mKate2-ras | 15 ng/µl | 37°C, 15 min | *Strzyz et al., 2015* |
| hsp70:Stathmin1-mKate2 | 15 ng/µl | 39°C, 10 min | *Taverna et al., 2016* |
| hsp70:mKate2-dynactin (1-811) | 15 ng/µl | 39°C, 15 min | *Taverna et al., 2016* |

## Morpholino experiments

To knockdown gene function, the following amounts of morpholinos (Gene Tools) were injected into the yolk at one cell stage: 2,5 ng of a splice morpholino for the sara gene (5'-TGAACTAGAGAC TTTACCTTGCCAC-3') (*Kressmann et al., 2015*), 2 ng of p53 MO, (5'-GCGCCATTGCTTTGCAA-GAATTG-3' [*Robu et al., 2007*]), 1 ng of standard control morpholino (5'-CCTCTTACCTCAGTTA-CAATTTATA-3'), 1 ng of a translation blocking morpholino for the notch1b gene (5'-GTATTGCATTC TCCTCTCCCGTCTG-3'), 0.41 ng of a translation blocking morpholino for the gene deltaC (MO1, 5'-AGCCATCTTTGCCTTCT TGTCTGCT-3') (*Zhang et al., 2008*), and 2 ng of another translation blocking morpholino for the gene deltaC (MO2, 5'-CGATAGCAGACTGTGAGAGTAGTCC-3') (*Holley et al., 2002*; *Okigawa et al., 2014*).

## Blastomere transplantation

Donor embryos from the double transgenic line Tg(hsp70:H2B-RFP),Tg(her4:EGFP) and acceptor were dechorionated in pronase. At the high stage, cells from the animal pole of donors were transplanted into the acceptor wild type embryos. Transplanted embryos were grown at 32°C to recover for 4 hours, then moved to 28°C. The E3 medium was supplemented with 100 U of penicillin and streptomycin (Thermo Fisher Scientific). Transplanted embryos were heat shocked at 24 hpf for 15 min at 39°C and imaged from 28 hpf.

## Drug treatments

Notch inhibitors DAPT (Sigma-Aldrich) and LY411575 (Sigma-Aldrich) were dissolved in DMSO and used at 50 and 10 µM concentration respectively and compared to DMSO only controls. Embryos were dechorionated and treated with the drugs at 28°C in E3 medium, either in a 24 multiwell plate or the light sheet microscope chamber for live imaging experiments. All treatments started at 24 hpf. For live imaging experiments, fish were preincubated with the drug for 4 hours and imaging started at 28 hpf.

## In vivo labelling of proliferating cells

To label proliferating cells in the Notch inhibition and Sara knock down experiments, the EdU Click-iT-Alexa 488 or 647 fluorophore kit (Invitrogen) was used. Embryos were incubated at 4°C in E3 medium supplemented with 500 µM of EdU in 10% DMSO for 1 hour before fixation. Embryos were then washed twice with E3 and fixed overnight in 4% PFA. After whole mount staining, EdU was detected using the Click-iT-Alexa 488 or 647 fluorophore kit (Invitrogen) according to manufacturer's protocol. Embryos were then stored in Phosphate buffered saline (PBS).

## Immunofluorescence

All immunostainings were performed on whole-mount embryos fixed in 4% paraformaldehyde (Sigma-Aldrich) in PBS as previously described (*Icha et al., 2016a*). In brief, embryos were permeabilised with trypsin, blocked with goat or donkey serum 10% in PBS-Triton 0.8% and incubated with the primary antibody for 72 hours. The Notch1b (GTX48505, GeneTex), DeltaC (zdc2, Abcam) and GFP (50430–2-AP, Proteintech) antibodies were used at 1:100 dilution. The acetylated tubulin antibody (T6793, Sigma-Aldrich) was used at 1:250 dilution. The phospho-histone H3 antibody (ab10543, abcam) was used at 1:500 dilution. Embryos were incubated for 48 hours with an appropriate fluorescently labeled secondary antibody (Molecular Probes) at 1:500 dilution and DAPI at 1:1000 dilution (Thermo Fisher Scientific).

## Image acquisition
### Laser scanning confocal microscopy

Fixed samples from immunostaining experiments were imaged with the Zeiss LSM 880 inverted point-scanning confocal microscope (Carl Zeiss Microscopy) using the 40x/1.2 C-Apochromat water immersion objective (Zeiss). Samples were mounted in 0.6% agarose in glass-bottom dishes (MatTek Corporation) and imaged at room temperature. The microscope was operated with the ZEN 2011 (black edition) software (Zeiss).

### Spinning disk confocal microscopy (SDCM)

Confocal stacks for fixed samples for Notch inhibition and Sara knockdown experiments were taken using an Andor SDCM system (Andor), consisting of an Andor IX 83 stand equipped with a CSU-W1 scan head (Yokogawa) with Borealis upgrade and an Andor iXon Ultra 888 Monochrome EMCCD camera. 40x/1.25 or 60x/1.3 silicone objectives were used to acquire 50–80 µm z-stacks were recorded with 1 µm optical sectioning. Fixed samples were mounted in 0.6% agarose in glass-bottom dishes (MatTek Corporation) and imaged at room temperature. The microscope was operated by Andor iQ 3.6 software.

### In vivo light sheet fluorescent imaging

Light sheet imaging started between 28 and 30 hpf. Embryos were manually dechorionated and mounted in glass capillaries in 0.6% low-melting-point agarose as previously described (*Icha et al., 2016b*). The sample chamber was filled with E3 medium containing 0.01% MS-222 (Sigma-Aldrich) to immobilise embryos and 0.2 mM N-phenylthiourea (Sigma-Aldrich) to prevent pigmentation. Imaging was performed on a Zeiss Lightsheet Z.1 microscope (Carl Zeiss Microscopy) equipped with two PCO.Edge 5.5 sCMOS cameras and using a 20x/1.2 Zeiss Plan-Apochromat water-immersion objective at 28.5°C. Z-stacks spanning the entire retinal epithelium (70–100 µm) were recorded with 1 µm optical sectioning every 5 min for 15–20 hr using the double-sided illumination mode. Sara endosomes inheritance during cell division was imaged with 1 min time resolution and 0.5 µm optical sectioning. The system was operated by the ZEN 2014 software (black edition).

## Image analysis

Images from live imaging experiments were cropped and averaged in ZEN Black and/or Fiji (*Schindelin et al., 2012*) and were corrected for drift using a Fiji plugin (Manual drift correction) created by Benoit Lombardot (Scientific Computing Facility, Max Planck Institute of Molecular Cell Biology and Genetics, Dresden, Germany).

### Analysis of sister cell somal translocation

Ath5 progenitor divisions were identified and cells were backtracked to the previous cell division in order to identify cells giving rise to Ath5+ progenitors and the sister cell arising from the division. From the point after division, the translocation of sister cells was tracked in 2D in maximum projected sub-stacks by following the center of the cell body in Fiji using the semi-automated ImageJ plugin MTrackJ (*Meijering et al., 2012*). MSDs and directionality ratios were calculated in the DiPer program (*Gorelik and Gautreau, 2014*), executed as a macro in Excel (Microsoft). To calculate the average instantaneous velocities, a custom-made script was written using Python 3 and can be found here (*Nerli, 2020*).

### Integral mean calculations

The integral mean $\hat{d}$ was computed using Python 3 (https://git.mpi-cbg.de/nerli/nerli-et-al-2020-scripts; *Nerli, 2020*) as follows:

$$\hat{d} = \frac{I}{T} \tag{1}$$

where $T$ is the time length of each cellular trajectory and I is the area below the trajectory curve, calculated as:

$$I = \int_0^T \text{depth}(t) > \mathrm{d}t \tag{2}$$

where depth($t$) >0.

### Retinal size measurements

Retinal size was manually measured using 48 hpf retinas treated with LY411575 10 µM or equal volume of DMSO, as illustrated in *Figure 1G*. Thickness and diameter were measured in the central z plane of the nasal, central and temporal region of the retina, using the Fiji line tool.

### Quantification of the number of EdU-positive cells

Prior to counting, a Gaussian blur of 0.5 was applied to all the images. The number of EdU positive cells was manually counted in the central part of the retinal neuroepithelium in Fiji, using a region of interest (ROI) that spans the entire length of the apicobasal axis (6400 $\mu m^2$). For each embryo, three Z slices per central region were analysed.

### Notch1b and DeltaC spatial distribution

Prior to analysis, a Gaussian blur of 0.5 was applied to all the images. Apical, central and basal ROIs of the same dimensions (560 $\mu m^2$) were set for each image, dividing the length of the apicobasal axis by 3. Three central slices per embryo were analysed in the nasal, central and temporal regions (*Figure 3A''*). SpotCounter plugin for Fiji (Spot counter, Nico Stuurman, ValelabUtils package) was used to measure the number of spots in each ROIs. The number of spots in each ROIs was then normalised by the total number of spots in the 3 ROIs of a slice.

### Sara-positive endosome count

Prior to analysis, a Gaussian blur of 0.5 was applied to all the images. Confocal z-stack of retinal progenitors were reconstructed in 3D using the volume renderer ClearVolume (*Royer et al., 2015*) for Fiji. The number of Sara-positive endosomes was manually counted in dividing cells from 3D reconstructions.

## Statistical analysis

All statistical tests used are indicated in the figure legend, as well as the definitions of error bars. All tests used were two sided and 95% confidence intervals were considered. P values and sample sizes are noted in figure panels or figure legends. Data was analysed using GraphPad Prism 6 or Python 3. Statistical analysis was performed using GraphPad Prism 6 and MATLAB.

## Acknowledgements

We thank the Norden Lab for fruitful project discussions. We are grateful to William A Harris, Julien Vermont and Iskra Yanakieva for the helpful comments on the manuscript. Christoph Zechner and Tommaso Bianucci are thanked for help with data analysis. We thank Sylvia Kaufmann, the Computer Department, Light Microscopy Facility, Scientific Computing, and Fish facility of the Max Planck Institute of Molecular Cell Biology and Genetics for experimental support. We also thank Marcos Gonzáles-Gaitán for sharing the CFP-Sara and mRFP-Sara constructs, Nikolay Ninov for sharing the Tp1bglob:H2BmCherry line, Nadine Vastenhow's lab, Wieland Huttner's lab and Michael Brand's lab for sharing probes and antibodies and help with experiments. EN is a member of the IMPRS-CellDevoSys PhD program and the IBB-Integrative Biology and Biomedicine PhD program and is supported by the MPI-CBG. CN is supported by MPI-CBG, the FCG-IGC, the German Research Foundation (NO 1069/5–1) and an ERC consolidator grant (H2020 ERC-2018-CoG-81904).

## Additional information

### Funding

| Funder | Grant reference number | Author |
|---|---|---|
| European Research Council | Consolidator grant H2020 ERC-2018-CoG-81904 | Caren Norden |
| Deutsche Forschungsgemeinschaft | NO 1068/5-1 | Caren Norden |
| Max-Planck-Gesellschaft | | Caren Norden |

The funders had no role in study design, data collection and interpretation, or the decision to submit the work for publication.

## Author contributions
Elisa Nerli, Conceptualization, Data curation, Formal analysis, Validation, Investigation, Visualization, Methodology, Writing - original draft; Mauricio Rocha-Martins, Conceptualization, Data curation, Supervision, Investigation, Methodology, Writing - original draft; Caren Norden, Conceptualization, Resources, Supervision, Funding acquisition, Writing - original draft, Project administration

## Author ORCIDs
Elisa Nerli (iD) https://orcid.org/0000-0003-4204-9702
Caren Norden (iD) https://orcid.org/0000-0001-8835-1451

## Ethics
Animal experimentation: All animal work in this study was performed in accordance with European Union directive 2010/63/EU, as well as the German Animal Welfare Act.

## Decision letter and Author response
Decision letter https://doi.org/10.7554/eLife.60462.sa1
Author response https://doi.org/10.7554/eLife.60462.sa2

# Additional files

## Supplementary files
• Transparent reporting form

## Data availability
All data generated or analysed during this study are included in the manuscript and supporting files. Source data files have been provided for all figures for which necessary.

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
