## [Decision Letter]

**Acceptance summary:**

This work nicely demonstrates the existence of asymmetric neurogenic divisions in the developing retina and rules out, in a clean and convincing way, previous hypotheses on graded Notch signaling along the apicobasal axis. It makes an important contribution to our understanding of retinal development.

**Decision letter after peer review:**

Thank you for submitting your article "Asymmetric neurogenic commitment of retinal progenitors is regulated via the Notch endocytic pathway" for consideration by *eLife*. Your article has been reviewed by two peer reviewers, one of whom is a member of our Board of Reviewing Editors, and the evaluation has been overseen by Marianne Bronner as the Senior Editor. The reviewers have opted to remain anonymous.

The reviewers have discussed the reviews with one another and the Reviewing Editor has drafted this decision to help you prepare a revised submission.

Summary:

The paper by Nerli et al. addresses the molecular control of the first neurogenic divisions in the developing zebrafish retina. Progenitors fated to become neurons express the transcription factor Ath5. Using Ath5-driven transgenic lines in live light-sheet microscopy to retrospectively track neurogenic progenitors, the authors demonstrate that these progenitors originate from divisions that are asymmetric in fate (one daughter cell becoming Ath5+) and in Notch signaling (the Ath5- daughter is Notch signaling high, as revealed using her4- or TP1-driven transgenes), the latter being necessary for the Ath5- fate. The remainder of the paper then aims to identify the mechanism by which a Notch signaling asymmetry is generated between daughter cells at the onset of neurogenic divisions. The work reported specifically addresses and challenges the conclusions made by previous studies that interkinetic nuclear migration combined with an apicobasal Notch gradient exposes progenitor nuclei to different Notch signaling levels. This hypothesis is ruled out here using a series of elegant experiments tracing individual nuclei and modifying their position by means of manipulating dynactin or stathmin. The authors then address the alternative hypothesis of intra-lineage signaling with an implication of Sara endosomes, as previously shown in *Drosophila* (Coumailleau et al., 2009) and zebrafish neural progenitors of the embryonic spinal cord (Kressmann et al., 2015). The authors observe (using overexpression of fluo-Sara mRNA) Sara endosomes carrying Notch1b and DeltaC proteins in retinal progenitors, and an asymmetric distribution of Sara endosomes in daughter cells. Knocking-down Sara (using a Sara MO) generates fewer neurons and an increased number of TP1-positive cells. The authors conclude that neurogenic asymmetry is driven by the asymmetric segregation of Sara endosomes, driving asymmetric Notch signaling in retinal progenitors.

This is a largely well-executed and nicely written piece of work. It convincingly rules out the previous hypothesis of apicobasal nuclear positioning, which had driven the field for years although solely based on in situ hybridization data. This in itself is important. However, the "positive" mechanistic data implying Sara endosomes remains correlative and describes a mechanism that has been previously reported. It is also at odds with previously published work (Kressmann et al., 2015). Key experiments directly implicating Sara endosomes in division asymmetry, asymmetrical Notch signaling and asymmetrical fate are lacking and should be provided to support the conclusions.

Essential revisions:

1) The authors report one case (=1 division) where Sara endosomes are concentrated in the Ath5- daughter cell. Attempts should be made to confirm this result. Live imaging may not be doable, but fixed material, where daughter cells are mosaically labeled could be analyzed for Ath5 and TP1 expression. This is crucial, as Sara is associated with the progenitor fate in the spinal cord (Kressmann et al), is postulated (but not shown) to have the same association here, but when depleted leads to more progenitors (Figure 5E, F and see point 3 below).

2) The effects of the Sara MO are suggestive of increased Notch signaling and symmetric proliferative, as opposed to asymmetric neurogenic, divisions. However, these hypotheses are not demonstrated directly. It could be possible, for example, that progenitors are blocked at the pre-division phase. The authors should directly show, using live cell tracking, that divisions become symmetric proliferative, generating two Ath5- (or TP1+) cells in the absence of Sara.

3) The results of the Sara MO experiments are strikingly different from those of Kressmann et al. In the present report, Sara knock-down leads to increased Notch signaling; in the spinal cord, Kressmann et al. showed that Sara knock-down led to decreased Notch signaling and a loss of the progenitor fate. Nerli et al. state "We propose that Sara endosomes are mostly inherited by the Ath5- cell, which retains high Notch signaling and thus stays in the proliferative state" and, under Sara knock-down: "there is an increase in active Notch signaling in both daughter cells as postulated by our working model". These two statements, which a appear contradictory, and the discrepancies with the work of Kressmann et al., should absolutely be discussed. What is the authors' hypothesis on how Sara endosomes would associate with the future progenitor cell and increased Notch signaling, but lead to more progenitor cells and more Notch when functionally depleted?

4) Figure 3: The Notch1b and DeltaC fluorescent antibody stainings have a lot of background and the brain is equally stained. Proper negative controls are needed for the Notch1b antibody. This means, controls should be performed using specimen lacking Notch1b (e.g. mutants or morphants), rather than specimen where the primary antibody is omitted (Figure 3—figure supplement 1). Also, could colorimetric in situs be performed? Given that the del Bene paper argued that it was notch1a, deltaB and deltaC RNA that was graded along the apicobasal axis of the retina, these in situ hybridization experiments should be repeated, in addition to the antibody stainings.

5) How do the authors reconcile that they did not observe graded DeltaC expression, whereas in the del Bene paper the RNA and antibody of DeltaC looks indeed graded? Murciano et al. (Figure 1C), also show clear Notch1 accumulation basally in the chicken retina, which should be discussed.

6) The author's state: “Heat shock expression of the hsp70:DNdynactin-mKate2 construct led to the translocation of both sister cell somas to greater and almost identical basal positions compared to controls. However, despite the fact that both sister cell somas reached very basal positions, only one became Ath5+.”

Murciano et al., also pharmacologically inhibited internuclear migration but saw and effect on neurogenesis. This finding should be discussed.

[Editors' note: further revisions were suggested prior to acceptance, as described below.]

Thank you for resubmitting your work entitled "Asymmetric neurogenic commitment of retinal progenitors is regulated via the Notch endocytic pathway" for further consideration by *eLife*. Your revised article has been evaluated by Marianne Bronner (Senior Editor) and a Reviewing Editor.

The manuscript has been significantly improved but there are some remaining issues that need to be addressed before acceptance, as outlined below:

Overall, this work is important in that it demonstrates the existence of asymmetric neurogenic divisions in the developing retina and rules out, in a clean and convincing way, previous hypotheses on graded Notch signaling along the apicobasal axis. The authors took the reviewer's comments to heart and adequately addressed critiques 1-6. However, the manuscript still shows a weakness in identifying the mechanism controlling this asymmetry. One key aspect that remains undemonstrated is the fact that Sara endosomes are involved in the asymmetry of progenitor division (requested revisions point 2). It is unclear why the authors state that live imaging is not possible to monitor symmetric divisions in Sara morphants, while the same method was used in the case of LY411575 application (Figure 1I). The absence of experiments showing that cell divisions become symmetric, diminishes the Sara-related message.

Overall, the reviewers are enthusiastic about the work but believe that in the absence of live imaging of cell divisions in Sara morphants, it is important not to overstate the function of Sara, irrespective of how logical it may seem.

The authors already changed the text to state that they speculate that cell divisions become symmetric after Sara MO but key titles and sentences were not correspondingly changed. For example: (i) title of paragraph "Asymmetric inheritance of Sara-positive endosomes affects the asymmetry of neurogenic commitment" – the MO manipulation does not affect the asymmetric inheritance of Sara endosome but their overall presence, and the effect on asymmetrical neurogenic commitment is not shown; (ii) title of Figure 5 "Sara-positive endosomes affect the asymmetry of divisions generating Ath5+ progenitors", and.… (iii) the title of the paper "Asymmetric neurogenic commitment of retinal progenitors is regulated via the Notch endocytic pathway".

---

## [Author Response]

Essential revisions:1) The authors report one case (=1 division) where Sara endosomes are concentrated in the Ath5- daughter cell. Attempts should be made to confirm this result. Live imaging may not be doable, but fixed material, where daughter cells are mosaically labeled could be analyzed for Ath5 and TP1 expression. This is crucial, as Sara is associated with the progenitor fate in the spinal cord (Kressmann et al), is postulated (but not shown) to have the same association here, but when depleted leads to more progenitors (Figure 5E,F and see point 3 below).

We now expanded our analysis of the lack of Tp1, Ath5 co-localization. We analyzed 215 cells expressing Ath5 from 4 different embryos and find that only in 5 cases weak expression of Tp1 with Ath5 is seen, while all other cells show no expression (text in subsection “Notch pathway components are uniformly distributed along the apicobasal axis of the retinal neuroepithelium” and Figure 3—figure supplement 2A, A’). This strongly argues that indeed Notch signaling is not active in cells starting to express Ath5 and our finding that Sara depletion increases the number of Tp1 positive cells and decreases the number of Ath5 positive cells argues that indeed Sara is responsible for a balanced distribution of this signaling pathway component (see also our interpretation on the Sara related pathway in point 3). In addition, we confirmed that Tp1 positive cells indeed carry Sara endosomes (Figure 3—figure supplement 2B).

2) The effects of the Sara MO are suggestive of increased Notch signaling and symmetric proliferative, as opposed to asymmetric neurogenic, divisions. However, these hypotheses are not demonstrated directly. It could be possible, for example, that progenitors are blocked at the pre-division phase. The authors should directly show, using live cell tracking, that divisions become symmetric proliferative, generating two Ath5- (or TP1+) cells in the absence of Sara.

This is an important point brought up by the reviewers and we previously had this idea ourselves. However, when we started to outline the actual experiments, we noted that we made a thinking mistake as this experiment is close to impossible to carry out. The reason is that at the stage at which the Ath5+ progenitors arise, most other progenitors still undergo symmetric proliferative divisions, generating two Ath5- cells, also in the control situation. Thus, finding symmetric proliferative Ath5- divisions would be the norm, not the exception, and it would not be feasible to detect an increase in symmetric proliferative divisions upon Sara knockdown with the cohort size that is possible in live-imaging experiments. We believe however that the experiments in response to Q1 and Q3 are convincing to substantiate our general hypothesis.

Nonetheless, to further underline this notion, we excluded the possibility that progenitors are stalled before or during division using pH3 staining. We find a similar amount of divisions in Sara morphants as in controls. This data in now shown in Figure 5—figure supplement 1B. Moreover, consistent with the hypothesis that Sara knockdown leads to more symmetric proliferative divisions, we demonstrate that morphant embryos feature a higher level of proliferation, an expected consequence for an increase in the number of symmetric proliferative divisions, shown by quantification of EdU labelling in controls versus Sara morphants (Figure 5—figure supplement 1B, C, D) at stages at which in control progenitors already leave the cell cycle and become Ath5+ (Figure 5—figure supplement 1C, control embryos, and Figure 1 E, F, control embryos). These experiments confirm that the balance between proliferation and differentiation is changed in Sara morphants.

Nevertheless, as we do understand the reviewer’s point, we tuned down our statement to “We thus speculate that depletion of Sara led to more symmetric proliferative divisions generating two Ath5- cells.”.

We also added a statement and compare our findings to previous findings and added a paragraph in the Discussion (see also point 3).

3) The results of the Sara MO experiments are strikingly different from those of Kressmann et al. In the present report, Sara knock-down leads to increased Notch signaling; in the spinal cord, Kressmann et al. showed that Sara knock-down led to decreased Notch signaling and a loss of the progenitor fate. Nerli et al. state "We propose that Sara endosomes are mostly inherited by the Ath5- cell, which retains high Notch signaling and thus stays in the proliferative state" and, under Sara knock-down: "there is an increase in active Notch signaling in both daughter cells as postulated by our working model". These two statements, which a appear contradictory, and the discrepancies with the work of Kressmann et al., should absolutely be discussed. What is the authors' hypothesis on how Sara endosomes would associate with the future progenitor cell and increased Notch signaling, but lead to more progenitor cells and more Notch when functionally depleted?

While these two statements might on first sight sound slightly counterintuitive, we do not believe that they contradict each other. Our interpretation is the following: In the control scenario, in which the Sara protein is expressed, Sara-positive endosomes are linked to Notch signaling and can be unequally distributed during cell divisions generating imbalances/asymmetries/differences in Notch activity among progenitor cells. While the cell inheriting the endosomes and high notch activity remains a progenitor, its sister cell is depleted of Notch and enters neurogenesis. However, when Sara is depleted from all the cells by morpholino knockdown, Notch continues to be active. In this case, Notch distribution between sister cells is not regulated any more through the endosomal pathway, so it cannot be depleted from one sister cell during division. Consequently, more cells maintain high Notch activity (Figure 5F), leading to increased proliferation and decreased neurogenesis (Figure 5—figure supplement 1C). This argues that Sara is responsible for controlled Notch distribution between sister cells.

Thus, the differences in outcome between retina and spinal cord imply that the fine-tuning of Notch depends on the specific neuronal system and lineage in which it is acting. One of these differences could be different commitment states in the lineages as seen between the spinal cord and the retina. The exact mechanisms of how this fine-tuning of Notch signaling through Sara-positive endosomes occurs will be interesting to explore in future studies.

We now discuss this point more explicitly in the Discussion.

4) Figure 3: The Notch1b and DeltaC fluorescent antibody stainings have a lot of background and the brain is equally stained. Proper negative controls are needed for the Notch1b antibody. This means, controls should be performed using specimen lacking Notch1b (e.g. mutants or morphants), rather than specimen where the primary antibody is omitted (Figure 3—figure supplement 1). Also, could colorimetric in situs be performed? Given that the del Bene paper argued that it was notch1a, deltaB and deltaC RNA that was graded along the apicobasal axis of the retina, these in situ hybridization experiments should be repeated, in addition to the antibody stainings.

We now added further controls and used the Notch1b and DeltaC antibodies on respective morphants (we used established morpholinos as the timeframe of the revisions and COVID situation did not allow for importing mutants).

In both cases, especially strong in the Notch scenario, we see disappearance or strong reduction of antibody staining compared to the control situation. Thus, we are confident that the antibody staining of controls is genuine in both scenarios (Figure 3—figure supplement 1A, B).

We further performed in situ hybridization for notch1a and deltaC, similar to experiments done in del Bene et al., 2008. We show the results in Author response image 1 as we do not believe that they significantly add to the manuscript. We note that our staining looks very similar to what was reported by del Bene (their Figure 5 A-C). However, as these are RNA probes, we believe that the antibody experiments are more meaningful as they reflect the protein localization. Despite the fact that there might be some graded RNA distribution, which is hard to tell at this resolution, our findings that there is no correlation between apicobasal nuclear positioning and neurogenic fate determination shows that such an RNA gradient, if existent, does not influence neurogenic potential.

**Author response image 1. sa2fig1:** In situ hybridization showing mRNA expression levels of notch1b (left) and deltaC (right). RNA probes targeting notch receptor 1b and deltaC were synthesized with digoxigenin label. Zebrafish embryos at 32 hpf were fixed overnight at 4 °C in PFA 4% (dissolved in PBS) and kept in 100% methanol at 20 °C until hybridization. The embryos were permeabilized with Proteinase K at room temperature. Hybridization was performed at 68 °C overnight, the probes were detected with anti-Dig antibody and alkaline phosphatase activity was revealed with BM purple.

5) How do the authors reconcile that they did not observe graded DeltaC expression, whereas in the del Bene paper the RNA and antibody of DeltaC looks indeed graded? Murciano et al. (Figure 1C), also show clear Notch1 accumulation basally in the chicken retina, which should be discussed.

See our response to point 4. Furthermore, in the Del Bene paper staining was performed using a different DeltaC antibody. We here used a DeltaC antibody that has been shown to be specific in zebrafish (see Wright et al., 2011 Development, Giudicelli, 2007 Plos Biology and others). It is not uncommon that RNA distribution and protein distribution slightly differ in tissues (shown in Spirov et al., 2009 for the bicoid mRNA and protein, and in Toyoda et al., 2010 for Fgf8 in the mouse telencephalon). However, as noted above, the fact that apicobasal nuclear positioning does not influence neurogenic fate determination shows that apicobasal graded distribution of Notch/Delta, if existent at the RNA level, is independent of neurogenic potential.

6) The author's state: “Heat shock expression of the hsp70:DNdynactin-mKate2 construct led to the translocation of both sister cell somas to greater and almost identical basal positions compared to controls. However, despite the fact that both sister cell somas reached very basal positions, only one became Ath5+.”Murciano et al., also pharmacologically inhibited internuclear migration but saw and effect on neurogenesis. This finding should be discussed.

It is hard to interpret the differences between our study and the previous study in which nuclear movement was blocked by a tissue wide cytochalasin B (CCB) treatment (Figure 5 in Murciano et al., 2002) due to the fact that CCB not only blocks IKNM (which was not explicitly shown as only fixed samples were presented) but also interferes with diverse other cellular processes such as cytokinesis (shown by the increase of mitotic figures in Figure 5B) and can lead to cell delamination (which could explain the non-apical mitotic cells observed in Figure 5B). It was not investigated which, if not all, of these effects apply in the previously presented scenario. Thus, it is hard to decipher in hindsight, the exact contributions that led to the increase in neurogenesis as division patterns (e.g. symmetric versus asymmetric) were not explored in this study.

In contrast, for our experiments we used genetic interference in single cells in an otherwise untreated neuroepithelium. Neither of the two independent genetic conditions, one that keeps cell soma at more apical positions and another one that sends cell soma more basally, had an influence on the asymmetric division patterns of cells producing one Ath5 positive sister cell. This fact makes us confident that the extent of basal migration does not influence the asymmetry of divisions leading to one neurogenic Ath5+ daughter cell.

We now added a brief statement discussing this item in the Discussion.

[Editors' note: further revisions were suggested prior to acceptance, as described below.]

Overall, the reviewers are enthusiastic about the work but believe that in the absence of live imaging of cell divisions in Sara morphants, it is important not to overstate the function of Sara, irrespective of how logical it may seem.The authors already changed the text to state that they speculate that cell divisions become symmetric after Sara MO but key titles and sentences were not correspondingly changed. For example: (i) title of paragraph "Asymmetric inheritance of Sara-positive endosomes affects the asymmetry of neurogenic commitment" – the MO manipulation does not affect the asymmetric inheritance of Sara endosome but their overall presence, and the effect on asymmetrical neurogenic commitment is not shown; (ii) title of Figure 5 "Sara-positive endosomes affect the asymmetry of divisions generating Ath5+ progenitors", and.… (iii) the title of the paper "Asymmetric neurogenic commitment of retinal progenitors is regulated via the Notch endocytic pathway".

In particular, we toned down the statements about the involvement of the Sara endocytic pathway in asymmetric neurogenic commitment. We changed:

– The title of the paper to “Asymmetric neurogenic commitment of retinal progenitors involves Notch through the endocytic pathway”;

– The title of the subsection to “Notch signalling affects neurogenic commitment of progenitors through Sara endosomes;”

– The title of Figure 5 to “Notch signalling affects neurogenic commitment of progenitors through Sara endosomes”.

We also like to take this opportunity to further clarify why we cannot monitor symmetric division with live imaging in Sara morphants as we did upon Notch inhibition. In the case of Notch inhibition using the LY411575 inhibitor, we showed that more symmetric neurogenic divisions occur, and more cells express Ath5 (Figure 1E, H). This experiment was possible as in this condition, both sister cells start expressing Ath5+, while in control conditions only one sister expresses Ath5 (Figure 1B) meaning we could use the onset of fluorescence as a readout. In Sara morphants however, fewer Ath5+ cells are present (Figure 5F) and we would expect to observe more symmetric proliferative divisions, in which both progenitors do not express Ath5.

However, as this type of symmetric proliferative divisions normally occurs also for the majority of cells in controls, we do not have a positive readout for cells that would have turned on Ath5 but do not do so in the Sara morphant condition. What we did show is an increase in the number of proliferative cells (Figure 5—figure supplement 1C) and a decrease in the number of Ath5+ cells (Figure 5F, Figure 5—figure supplement 1C), which in our opinion can only be explained by an increase of symmetric proliferative divisions generating two Ath5- progenitors at the expenses of asymmetric neurogenic divisions. We hope this clarifies this issue, but we agree that tuning down our statements further improves the manuscript.